# Influence of biomass burning vapor wall loss correction on modeling organic aerosols in Europe by CAMx v6.50

Jianhui Jiang[1], Imad El Haddad[1], Sebnem Aksoyoglu[1], Giulia Stefenelli[1], Amelie Bertrand[1,2], Nicolas Marchand[2], Francesco Canonaco[1], Jean-Eudes Petit[3], Olivier Favez[3], Stefania Gilardoni[4], Urs Baltensperger[1], André S. H. Prévôt[1]

[1]Laboratory of Atmospheric Chemistry, Paul Scherrer Institute, 5232 Villigen PSI, Switzerland
[2]Aix Marseille Univ, CNRS, LCE, Marseille, France
[3]Institut National de l'Environnement Industriel et des Risques (INERIS), Verneuil-en-Halatte, France
[4]Italian National Research Council – Institute of Polar Sciences, Bologna, Italy

*Correspondence to*: Imad El Haddad (imad.el-haddad@psi.ch), Sebnem Aksoyoglu (sebnem.aksoyoglu@psi.ch) and Jianhui Jiang (jianhui.jiang@psi.ch)

**Abstract.** Increasing evidence from experimental studies suggests that the losses of semi-volatile vapors to the chamber walls could be responsible for the underestimation of organic aerosol (OA) in air quality models which use parameters obtained from the chamber experiments. In this study, a box model with volatility basis set (VBS) scheme was developed and the secondary organic aerosol (SOA) yields with vapor wall loss corrected were optimized by a genetic algorithm based on advanced chamber experimental data for biomass burning. The vapor wall loss correction increases the SOA yields by a factor of 1.9–4.9, and leads to a better agreement with the measured OA for 14 chamber experiments under different temperatures and emission loads. To investigate the influence of vapor wall loss correction on regional OA simulations, the optimized parameterizations (SOA yields, emissions of intermediate-volatility organic compounds from biomass burning, and enthalpy of vaporization) were implemented in the regional air quality model CAMx (Comprehensive Air Quality Model with extensions). The modeled results from the VBS schemes with standard (VBS_BASE) and vapor wall loss corrected parameters (VBS_WLS), as well as the traditional two-product approach were compared and evaluated by OA measurements from five Aerodyne aerosol chemical speciation monitor (ACSM)/aerosol mass spectrometer (AMS) stations in the winter of 2011. An additional reference scenario VBS_noWLS was also developed using the same parameterization as VBS_WLS except for the SOA yields which was optimized assuming there is no vapor wall loss. The VBS_WLS generally shows the best performance for predicting OA among all OA schemes, and reduces the mean fractional bias from -72.9% (VBS_BASE) to -1.6% for the winter OA. In Europe, the VBS_WLS produces the highest domain average OA in winter (2.3 µg m$^{-3}$), which is 106.6% and 26.2% higher than VBS_BASE and VBS_noWLS, respectively. Compared to VBS_noWLS, VBS_WLS leads to an increase in SOA by up to ~80% (in Balkans). VBS_WLS also leads to a better agreement between the modeled SOA fraction in OA (fSOA) and the estimated measured values in the literature. The substantial influence of vapor wall loss correction on modeled OA in Europe highlights the importance of further improvements in the parameterizations based on laboratory studies with a wider range of chamber conditions and field observations with higher spatial and temporal coverage.

# 1 Introduction

Organic aerosol (OA) accounts for a substantial fraction of atmospheric particulate matter (Jimenez et al., 2009), which is closely associated with human health impacts and climate change (Cohen et al., 2017; Kanakidou et al., 2005; Lelieveld et al., 2015). Organic aerosol originates from a variety of natural and anthropogenic sources (Hallquist et al., 2009), among which residential biomass burning emission has been recognized as the dominant source for both primary (POA) and secondary (SOA) organic aerosols in Europe during winter time (Butt et al., 2016; Jiang et al., 2019b; Qi et al., 2019). Despite its substantial

contribution to OA, biomass burning OA is largely underestimated by chemical transport models (CTM) (Ciarelli et al., 2017a; Hallquist et al., 2009; Robinson et al., 2007; Theodoritsi and Pandis, 2019; Woody et al., 2016).

Many efforts have been devoted to understand and diminish the gap between modeled and observed OA from biomass burning. One of the major reasons of the underestimated OA is the absence of semi-volatile organic compounds (SVOCs) from residential biomass burning in the current emission inventories (Denier van der Gon et al., 2015). A smog-chamber study

showed that the precursors traditionally included in the CTMs account for only ~3-27% of the observed SOA from residential biomass burning (Bruns et al., 2016). In order to compensate the effects from missing precursors, various modeling studies treated the POA as semi-volatile and adopted different scaling approaches to calculate the S/IVOC emissions. The most commonly used method is to increase the POA emissions by a factor of 3 (Ciarelli et al., 2017a; Fountoukis et al., 2014; Jiang et al., 2019b; Tsimpidi et al., 2010), while recent studies have also developed new profiles based on the nonmethane organic

compounds (NMOCs) (Cai et al., 2019; Lu et al., 2018). However, increasing number of laboratory experimental studies found that the S/IVOC emissions are of high variability depending on different burning conditions and fuel types (Hatch et al., 2015; Hatch et al., 2017; Jen et al., 2019; Koss et al., 2018; Sekimoto et al., 2018), and the estimation of S/IVOC in modeling studies remains to be improved. Meanwhile, increasing evidence from chamber experiments demonstrated that the losses of semi-volatile vapors to the chamber walls could lead to a substantial underestimation of OA (Akherati et al., 2020; Bertrand et al.,

2018; Bian et al., 2015; Krechmer et al., 2016; Loza et al., 2010; Matsunaga and Ziemann, 2010; Zhang et al., 2014). Unlike the particle wall losses – which have been routinely corrected in the chamber studies – the effects of vapor wall losses are rarely investigated and considered in modeling practices.

Zhang et al. (2014) reported that the vapor wall losses may lead to an underestimation of SOA by a factor of 1.1-4.2, depending on different $NO_x$ conditions. This factor has been adopted by several CTM studies to scale the yields of SOA up.

For instance, Baker et al. (2015) tested the sensitivity of CMAQ to the vapor wall loss by increasing the yields of semi-volatile gases by a factor of 4, in which the traditional two-product approach was used for OA simulations. This factor was also implemented in a box model with volatility basis set (VBS) scheme (Hayes et al., 2015), which distributed organic species into logarithmically spaced volatility bins and was shown to improve the model performance for predicting SOA (Donahue et al., 2011; Donahue et al., 2006; Hodzic et al., 2010; Robinson et al., 2007). Nevertheless, recent studies showed that the vapor

wall losses lead to even larger variability on SOA yields according to different chamber conditions and precursor species (Akherati et al., 2019; Cappa et al., 2016). On the other hand, some studies took the vapor wall loss corrections into account

in CTMs using the SOA yields generated from the Statistical Oxidation Model (SOM). Cappa et al. (2016) and Akherati et al. (2019) used the traditional two-product model to fit vapor wall loss corrected SOA yields, and applied the yields in the regional CTM UCD/CIT. They reported, however, that the two-product fits might not be sufficiently robust. Furthermore, Hodzic et al. (2016) used a vapor wall loss corrected VBS parameterization in the global model GEOS-Chem based on chamber experiments conducted on individual precursors, which are highly dependent on the experimental conditions. Each of these latter studies clearly called for a better assessment of the uncertainties across the entire range of precursor compounds as well as under different chamber conditions.

Here, we 1) developed a VBS-based box model and fit the vapor wall loss corrected SOA yields of biomass burning IVOC based on the 14 chamber experiments under different temperature and emission loads, 2) implemented the vapor wall loss corrected VBS parameters in the regional chemical transport model Comprehensive Air Quality Model with extensions (CAMx), and 3) investigated the role of vapor wall loss correction on model performance by comparing modeled organic aerosols from traditional and modified VBS OA schemes with ambient observations at multiple European sites. The biomass burning in this study refers to residential biomass burning, while the wildfires and prescribed burning are not included.

## 2. Parameterization method

### 2.1 Chamber experimental data

The parameterization of the VBS scheme was based on experimental data from two smog chamber campaigns in 2014-2015. It includes 14 experiments conducted under various temperature conditions (-10ºC, 2ºC, 15ºC) and covered a wide range of emission loads (from 19 to 284 µg m$^{-3}$). Emissions were generated by combustion of beech wood in three different wood stoves including conventional and modern burners manufactured in 2002-2010. Beech wood is selected as it is one of the major forest types in Europe, and beech wood is widely used for residential heating and cooking in Europe. Although different biomass fuel types may largely affect the emitted organic gas species and affect the SOA formation, a recent study showed that the effect of biomass fuel type on SOA formation is much smaller than the effects of initial OM load and OH exposure (Lim et al., 2019). The organic gases covering 86 intermediate-volatility and semi-volatile organic compounds (S/IVOC) which are SOA precursors, were measured by a proton-transfer-reaction time-of-flight mass spectrometer (PTR-ToF-MS). The PTR-ToF-MS was operated under standard conditions in H$_3$O$^+$ mode, as introduced in Stefenelli et al. (2019). A common set of 263 ions was extracted from the measurements, and among these ions, 86 showed clear decay with time and were identified as potential SOA precursors. These are listed in Table S1 of Stefenelli et al. (2019). The aerosol evolution was monitored by a high-resolution time-of-flight aerosol mass spectrometer (HR-ToF-AMS). The particle wall loss has already been corrected as described in Stefenelli et al. (2019). The conditions of each chamber experiment are shown in Table S1. More detailed description of the experiments can be found in Stefenelli et al. (2019), Bertrand et al. (2017) and Bruns et al. (2016).

## 2.2 VBS box model

A VBS box model was developed to simulate the formation and evolution of primary and secondary OA in the chamber. In the model, we assumed that the condensable gases generated from oxidation of the precursors could 1) partition to the particle phase, 2) be lost on the chamber wall, as well as 3) be diluted by other gases injected into the smog chamber. CAMx includes four types of precursors from anthropogenic sources, i.e. toluene, xylene, benzene, and IVOC which includes all the other unspeciated organic gases. According to our measurements, the traditional anthropogenic precursors toluene, xylene and benzene only account for ~15% of the total organic gases. To facilitate the implementation of the optimized parameters in CAMx, all the measured SOA precursors including the traditional ones were lumped into one surrogate as IVOC with the same reaction rate and volatility distribution. In comparison, Stefenelli et al. (2019) have assigned the same set of compounds to six different classes according to their properties (reaction rates, expected SOA yields…) and origins/occurrence in the emissions. These included furans and methoxy-phenols from the pyrolysis of cellulose and lignin, respectively, single-ring and poly-aromatic hydrocarbons from flaming combustion, and oxygenated non-aromatic compounds with lower and higher than six carbon chains. The current lumping approach of all these species into one surrogate, despite variations in their properties is more adapted for the implementation into CAMx and for assessing vapor wall losses, where additional parameters are included in the box model. The organic compounds were distributed into 6 logarithmically spaced volatility bins, corresponding to saturation concentrations of $10^{-1}$, $10^0$, $10^1$, $10^2$, $10^3$, and $10^4$ µg m$^{-3}$. The change in the organic gas concentration ($C$) for a constituent within the volatility bin $i$ ($C_i$) can be described by Eq. (1), where $P$ is the production of organic gas (OG) in the chamber due to oxidization of precursors, $k_{cs}$ is the condensation sink (s$^{-1}$) describing the speed of condensable gases condensing on existing aerosol particles, $k_w$ is the rate constant of vapor lost to the wall, $k_{dil}$ is the dilution rate, and $Ceq_{i,p}$ and $Ceq_{i,w}$ represent the gas-phase equilibrium concentrations to the aerosol particles and chamber wall surface, respectively:

$$\frac{dC_i}{dt} = P \cdot \zeta_i - k_{cs}\left(C_i - Ceq_{i,p}\right)$$
$$- k_w\left(C_i - Ceq_{i,w}\right) - k_{dil} \cdot C_i \tag{1}$$

The production rates of oxidized organic gases (P) are used as inputs of the box model. It is determined by the consumption rates of precursors measured by PTR taking into account their dilution. $\zeta_i$ represents the mass fraction of primary and oxidation products in a volatility bin $i$. $\zeta_i$ of POA from biomass burning is obtained from May et al. (2013), with values of 0.2, 0.1, 0.1, 0.2, 0.1, 0.3 for compounds in each volatility bin. $\zeta_i$ of oxidation products are assumed to follow a kernel normal distribution as a function of $logC^*$, $\zeta \sim N(\mu, \sigma^2)$, where $\mu$ is the median value of $logC^*$ and $\sigma$ is the standard deviation, which will be optimized as described in Section 2.3. The assumption of normal distribution could ensure positive $\zeta_i$ values, allow constraining the total mass fraction of the certain surrogate equals 1, and reduce the model's degree of freedom significantly, as reported in Stefenelli et al. (2019). The time series of $k_{dil}$ is obtained from Stefenelli et al. (2019). The $k_{cs}$ of each experiment is obtained from Bertrand et al. (2018). The $k_w$ varies significantly depending on the chamber conditions such as the chamber size, relative humidity, *etc*. Zhang et al. (2014) reported $k_w$ values of $2.5 \times 10^{-4}$ s$^{-1}$ and $1 \times 10^{-4}$ s$^{-1}$ for toluene and other VOCs respectively, while it is much higher in recent studies such as $1.2 \times 10^{-3}$ to $2.4 \times 10^{-3}$ s$^{-1}$ in Krechmer et al. (2016), $1.28 \times 10^{-3}$ s$^{-1}$

in Akherati et al. (2020), and ~$1\times10^{-3}$ to $3.3\times10^{-3}$ $s^{-1}$ in Bertrand et al. (2018). To cover the wide range of vapor wall loss, we tested three $k_w$ values 0.0020 $s^{-1}$, 0.0033 $s^{-1}$, 0.0040 $s^{-1}$ based on the condition of our chamber. A base case was also developed assuming there is no vapor wall loss in the chamber ($k_w = 0$). The condensation of a species in the particle-phase ($C_p$) can then be described by Eq. (2).

$$\frac{dC_{i,p}}{dt} = k_{cs}\left(C_i - Ceq_{i,p}\right) - k_{dil} \cdot C_{i,p} \tag{2}$$

Following the partitioning model of Pankow (1994), the gas-phase concentrations at equilibrium with respect to the particle phase ($Ceq_{i,p}$) and to the chamber wall ($Ceq_{i,w}$) are determined by their partitioning coefficients $\xi_i$ and $\xi_{i,w}$ (Donahue et al., 2009), as shown in Eq. (3) and Eq. (4):

$$Ceq_{i,p} = (C_{i,g} + C_{i,p}) \cdot [1 - \xi_i], \quad \xi_i = \left(1 + \frac{c_i^*}{c_{OA}}\right)^{-1} \tag{3}$$

$$Ceq_{i,w} = (C_{i,g} + C_{i,w}) \cdot [1 - \xi_{i,w}], \quad \xi_{i,w} = \left(1 + \frac{c_i^*}{c_{wall}}\right)^{-1} \tag{4}$$

where $C^*$ represents the saturation concentration, $C_{OA}$ is the wall-loss corrected OA concentration measured by the AMS, $C_{wall}$ is the equivalent organic mass concentration at the wall determined in Bertrand et al. (2018). The Clausius-Clapeyron equation (Eq. 5) was applied to take into account the effects of temperature on $C^*$:

$$C^* = C_{T0}^* \cdot \frac{T_0}{T} \cdot exp\left(\frac{\Delta H_{vap}/8.314}{1/T_0 - 1/T}\right) \tag{5}$$

where $C_{T0}^*$ is the mass saturation concentration under the reference temperature ($T_0$). $T$ is the temperature of each experiment,
while $T_0$ equals 298 K. $\Delta H_{vap}$ (J) is the enthalpy of vaporization at reference temperature, and 8.314 is the universal gas constant ($J$ $mol^{-1}$ $K^{-1}$). $\Delta H_{vap} = \{70000 - 11000 \times \log C^*\}$ is adopted for the primary set (May et al., 2013), while $\Delta H_{vap}$ of the oxidized products is determined during model optimization. The $C_{wall}$ was determined in previous studies as on the order of a few mg $m^{-3}$ (Bertrand et al., 2018). In this study, we run the box model for three different $C_{wall}$ values (1, 5, 25 mg $m^{-3}$) with a reference temperature of 2 °C (275.15 K) according to Bertrand et al. (2018).

## 2.3 Model optimization


The model is optimized to constrain the volatility distribution (as a function of $logC^*$, $\zeta \sim N(\mu, \sigma^2)$) and $\Delta H_{vap}$ of the oxidized products. A genetic algorithm (GA) is used to find the best-fit parameters leading to the lowest average root-mean-square error (RMSE) and mean bias (MB) between modeled and measured OA concentrations for all 14 experiments. The genetic algorithm is a metaheuristic algorithm inspired by the natural selection process to generate optimized solutions (Mitchell, 1996). It begins
by creating an initial population of individual solutions (20 different combinations of $\mu$, $\sigma$, $\Delta H_{vap}$ here) within certain upper and lower bounds, as called parents. The performance of each solution is evaluated by a fitness function, which is the sum of RMSE and MB between modeled and measured OA concentrations of 14 experiments in this study. A new generation of solutions is then formed either by making random changes to a single parent (called mutation) or by combining the vector entries of a pair of parents (called crossover). The process will be repeated until reaching the stopping conditions, which are

either the iterations time reaching 50 or the stall generations (generation with no significant change of fitness function) reaching 20. The GA is conducted using the genetic algorithm solver of Global Optimization Toolbox of MATLAB R2019a (The MathWorks, Inc).

## 3. Modeling approach

### 3.1 Regional chemical transport model CAMx

The regional model CAMx version 6.50 (Ramboll, 2018) was used to model organic aerosol in Europe (15ºW – 35ºE, 35ºN – 70ºN) for the whole year of 2011, with a horizontal resolution of 0.25º × 0.125º and 14 terrain following vertical layers from ~20 m above ground reaching up to 460 hPa. The Carbon Bond 6 Revision 2 (CB6r2) gas-phase mechanism (Hildebrandt Ruiz and Yarwood, 2013) was selected. The gas-aerosol partitioning of inorganic aerosols was simulated by the ISORROPIA thermodynamic model (Nenes et al., 1998). For organic aerosols, several OA schemes including both the traditional 2-product

approach (SOA chemistry/partitioning scheme, SOAP) and the VBS scheme with different parameterizations were applied (see Section 3.2).

The meteorological parameters were prepared with the Weather Research and Forecasting model (WRF, version 3.7.1; Skamarock et al., 2008) based on the 6-h European Centre for Medium–Range Weather Forecasts (ECMWF) reanalysis global data (Dee et al., 2011). The meteorological parameters were evaluated and reported in a previous study (Jiang et al., 2019a),

which showed that most of the meteorological parameters met the criteria for meteorological model performance by Emery (2001). The initial and boundary conditions were obtained from the global model MOZART-4/GEOS-5 (Horowitz et al., 2003). Inputs of ozone column densities were produced based on the Total Ozone Mapping Spectrometer (TOMS) data by the National Aeronautics and Space Administration (NASA, ftp://toms.gsfc.nasa.gov/pub/omi/data/), and the photolysis rates were then calculated by the Tropospheric Ultraviolet and Visible (TUV) Radiation Model version 4.8 (NCAR, 2011). The

source specific anthropogenic emissions were based on the European emission inventory TNO-MACC (Monitoring Atmospheric Composition and Climate)-III (Kuenen et al., 2014). The biogenic emissions (isoprene, monoterpenes, sesquiterpenes, soil NO) were simulated by the PSI model developed at the Laboratory of Atmospheric Chemistry at the Paul Scherrer Institute (Andreani-Aksoyoglu and Keller, 1995; Jiang et al., 2019a; Oderbolz et al., 2013). More details about the model inputs can be found in our previous studies performed using the same input data (Jiang et al., 2019a; Jiang et al., 2019b).

### 3.2 Parameterization of OA schemes

To investigate the effects of vapor wall loss corrected yields, as well as to compare to other modifications/parameterizations that are currently strongly debated in the community, five simulations with different OA schemes were conducted in this study (Table 1). Besides VBS_WLS which uses the optimized parameterization with vapor wall loss correction for the biomass burning sector, SOAP and VBS_BASE represent the two standard parameterization in CAMx; VBS_3POA represents a

common approach to offset the missing SVOC emissions in recent modelling studies without vapor wall loss; VBS_noWLS

is another reference case for that without vapor wall loss, which uses exactly the same parameters as VBS_WLS except for the SOA yields from IVOCs. Details about each OA schemes are introduced below:

- *SOAP*. The SOAP (SOA chemistry/partitioning) module is a semi-volatile equilibrium scheme based on the traditional two-product approach. The POA emissions are assumed to be inert in SOAP. The updated parameterization of SOAP2.1 in CAMx v6.50 used the aerosol yield data that correct for vapor wall losses in smog chamber experiments based on Zhang et al. (2014).

- *VBS_BASE*. The VBS_BASE used the standard VBS parameterization in CAMx v6.50. The IVOC emissions from different sources were calculated based on literature. The IVOCs from gasoline and diesel vehicles were calculated as 25% and 20% of NMVOC emissions from gasoline and diesel vehicles, respectively (Jathar et al., 2014). IVOC emissions from residential biomass burning were estimated as 4.5 times of POA emissions based on Ciarelli et al. (2017a). The IVOC emissions from other anthropogenic sources were calculated as 1.5 times of POA as proposed by Robinson et al. (2007).

- *VBS_3POA*. An increasing number of experimental and modeling studies have reported a considerable contribution of semi-volatile organic compounds (SVOCs) to SOA formation (Bruns et al., 2016; Ciarelli et al., 2017b; Denier van der Gon et al., 2015; Hatch et al., 2017; Woody et al., 2015), while the SVOCs are absent in the current emission inventories. Despite considerable variability of the SVOC emissions from biomass burning according to recent studies, the VBS_3POA is supposed to be a reference case representing the commonly used approach without vapor wall loss, and therefore we adopted the routine approach of multiplying the POA emissions by a factor of 3 to offset the influence of missing SVOC emissions. This approach has been widely used in modeling studies (Ciarelli et al., 2016; Ciarelli et al., 2017a; Shrivastava et al., 2011; Tsimpidi et al., 2010). All the other parameters were kept the same as the standard VBS parameterization in CAMx v6.50. The VBS_BASE IVOC emissions were adopted here.

- *VBS_WLS*. The VBS_WLS used the optimized parameters by the VBS box model, including the emissions and vapor wall loss corrected yields for IVOC from residential biomass burning, and the $\Delta H_{vap}$ of the oxidized products. The modified parameters for volatility bin specific yields and $\Delta H_{vap}$ of the oxidized products from IVOC can be found at https://doi.org/10.5281/zenodo.3998342. The optimized mass yields in the box model were converted to molar yields using the default molecular weights in CAMx (Table 1). Both the optimized and default molar yields have a sum larger than 1 as the VBS scheme accounts for both oxygenation and fragmentation (Koo et al., 2014). The reaction rate with OH ($k_{OH}$) was calculated based on the measurements following Stefenelli et al. (2019). Based on the chamber measurements, the IVOC emissions from residential biomass burning is ~13.7 times the primary OM load (Fig. S1), among which the traditional precursors in CAMx from biomass burning (toluene, xylene and benzene) accounting for ~15% of the total emissions. To avoid double counting of these traditional precursors which are already included in the emission inventory, we applied a factor of 12 to calculate the IVOC emissions from biomass burning. The IVOC emissions from other sources were estimated by the same approach as in VBS_3POA.

- *VBS_noWLS.* The VBS_noWLS was designed as a reference of VBS_WLS, which adopted the same parameters as VBS_WLS except for the yields. The VBS_noWLS used the fitted yields from the box model assuming that there is no vapor wall loss ($k_w = 0$).

## 3.3 Model evaluation

The general model performance for the major air pollutants ($SO_2$, $NO_2$, $O_3$, $PM_{2.5}$) was reported in our previous study (Jiang et al., 2019b), which was comparable to other modeling studies in Europe. OA measurements and source apportionment studies using positive matrix factorization (PMF) analysis from five Aerodyne aerosol chemical speciation monitor (ACSM) /aerosol mass spectrometer (AMS) stations in winter of 2011 were used to evaluate the modeled primary and secondary organic aerosol by different OA schemes: Zurich (Canonaco et al., 2013), Marseille (Bozzetti et al., 2017), SIRTA (Site Instrumental de Recherche par Télédétection Atmosphérique) facility located in the Paris region (Zhang et al., 2019), as well as Bologna and San Pietro Capofiume (SPC) (Paglione et al., 2020). For Zurich and SIRTA, only data collected from late autumn to early spring (January, February, March, November and December) - when emissions from biomass burning are relatively high - were used for the statistical analysis, although the observations covered longer time periods. The spatial distribution and observation periods of each station are shown in Fig. S2. The statistical metrics including mean bias (MB), mean error (ME), root-mean-square error (RMSE), mean fractional bias (MFB) and mean fractional error (MFE) between modeled and observed primary and secondary OA were calculated.

## 4. Results and discussion

### 4.1 Modeled and measured OA from chamber experiments

The optimized parameters were then applied to the box model to simulate OA production for 14 chamber experiments. Figure 1 shows the comparison between measured OA and modeled primary and secondary OA under the median chamber conditions ($k_w = 0.0033$ s$^{-1}$, $C_{wall} = 5$ mg m$^{-3}$) for each experiment. The model reproduces the process of OA formation for most of the experiments well, except for experiment #9 and #14 which have relatively lower OM loads (26 and 48 μg m$^{-3}$ for Exp9 and Exp14, respectively). It can be partially explained by different weighting impact for experiments with high or low OM loads. The experiments with higher OM loads normally have larger MB and RMSE in the beginning of optimization, and therefore have higher impact during the model optimization. A direct consequence is the optimized parameters would work better for those experiments with higher OM loads. However, the model performance on each experiment could also be influenced by a series of other factors such as temperature and chamber conditions. While the model simulation without vapor wall loss correction largely overestimates the OA at the initial time point and underestimates the final OA (Fig. S3), the agreement between the modeled and measured trends was improved when the vapor wall loss is taken into account. The mean bias (MB) and root mean square error (RMSE) between the modeled and observed OA in 14 experiments are 6.7 μg m$^{-3}$ and 42.2 μg m$^{-3}$ for the case under median chamber conditions, which are 48% and 12% lower than in the case without vapor loss correction

(MB = -12.8 µg m$^{-3}$, 47.8 µg m$^{-3}$). To investigate the role of vapor wall loss on the modeled OA, another set of simulations were performed, in which we used the same optimized parameterization under the median chamber conditions but set $k_w = 0$. In these cases, the modeled OA concentrations (dashed line in Fig. 1) were based on the assumption that there is no vapor wall loss. The wall loss ratio $R_{wall}$, which is defined as the ratio between modeled OA concentration without ($k_w = 0$) and with ($k_w = 0.0033$ s$^{-1}$, $C_{wall} = 5$ mg m$^{-3}$) vapor wall loss, was calculated for the endpoint of each experiment (Fig. 1c). The $R_{wall}$ values varied from 1.5 (Exp2) to 3.2 (Exp11) among the 14 experiments, and showed a clear dependence on the initial OA loads.

To further understand the factors influencing $R_{wall}$, we conducted a series of model simulations with and without vapor wall loss under different initial organic mass load, temperature and condensation sink inputs (Fig. 2). Higher $k_w$ and $C_{wall}$ lead to higher $R_{wall}$ values for all the cases, and different chamber conditions ($k_w$, $C_w$) could result in a different $R_{wall}$ by a factor of 1.2–1.6, depending on different temperature, OM loads and condensation sinks. The $R_{wall}$ values generally decrease with increasing initial OM loads, which is consistent with the fact that $R_{wall}$ values for Exp8–14 are higher than Exp1–7. The increased $R_{wall}$ with the increasing temperature explains why the Exp10–14 (T = 15 ºC) have higher $R_{wall}$ than Exp8 and Exp9 (T = –10 ºC) while they have similar OM load levels. The condensation sink is inversely correlated with $R_{wall}$, indicating that the higher the rate of condensable gases condensing on the existing particles, the lower the vapor loss to the chamber wall, and therefore the lower the effect of vapor wall loss on modeled OA.

The optimized volatility distribution for the secondary condensable gases from biomass burning (ppm per ppm IVOC) based on different wall loss assumptions ($k_w>0$ or $k_w=0$) are displayed in Fig. 3a. The optimized yields considering the vapor wall loss leads to a 3.3 times higher mass in the low-volatility bins ($logC^* \leq 0$) compared to that assuming $k_w=0$, indicating significant effects of vapor wall loss correction on predicting the SOA production. To give a more direct view about the effects of vapor wall loss on the SOA yield, we integrated the mass of SOA for all the volatility bins at 298 K (Fig. 3b). The mass yield under the median chamber conditions for vapor wall loss ($k_w = 0.0033$ s$^{-1}$, $C_{wall} = 5$ mg m$^{-3}$) is higher than the base case without considering the vapor wall loss about by factors of 4.9 (when $C_{OA} = 0.1$ µg m$^{-3}$) to 1.9 (when $C_{OA} = 1000$ µg m$^{-3}$). The influence of vapor wall loss on mass yield decreases with decreasing temperature. At 0 ºC, the mass yield with vapor wall loss correction is higher than the base case by factors of 4.3 (when $C_{OA} = 0.1$ µg m$^{-3}$) to 1.7 (when $C_{OA} = 1000$ µg m$^{-3}$).

## 4.2 Performance of CAMx with different OA schemes

The modeled OA concentrations with different OA schemes were compared with measurements from five ACSM/AMS stations in winter. The statistical results are shown in Table 2, and the distributions of OA concentrations and mean bias between modeled and measured primary and secondary OA are displayed in Fig. 4. The OA are overall underestimated with all OA schemes. The VBS schemes lead to a better model performance than the two-product approach SOAP, except for VBS_BASE with default VBS parameterization. These results are consistent with a previous study using CAMx (Meroni et al., 2017), in which the better performance of SOAP compared to the default VBS was reported as a result of error compensation. The improved performance of modified VBS (3POA, noWLS, WLS) for OA mainly comes from the contribution of SOA (Table 2). The modeled SOA by 3POA and noWLS are very similar, therefore the analysis below will

focus on the comparison between noWLS and WLS, for which the only difference is that WLS uses vapor wall loss corrected yields for IVOCs from biomass burning while noWLS uses the fitted yields assuming no vapor wall loss ($k_w$=0). WLS reduces the MFB between the modeled and measured SOA from 52.5% in noWLS to 20.0%. WLS shows a better average MB than noWLS, however, also increases the upper whisker of the MB (Fig. 4b), largely affected by overestimated SOA in Bologna and SPC.

Limited by the availability of OA measurements, the effects of vapor wall loss correction on model performance present a clear site dependence in this study. The modeled and measured daily average OA concentrations at each site are shown in Fig. 5. The temporal variations of primary and secondary OA at these sites can be found in Fig. S4. VBS_WLS leads to the best performance for both OA and SOA in Marseille and SIRTA, in spite of an overall underestimation (Fig. S4b, c). In Zurich, the vapor wall loss corrected yields for biomass burning improve the model performance in February and March, while there is an overestimation of the OA and SOA for all the OA schemes in November (Fig. S4a). The largest contribution to OA during this period was found to be from the biogenic SOA, which was potentially overestimated due to the overestimated temperatures during the same time period (Jiang et al., 2019b). Bologna and SPC are located in the Po Valley where biomass burning contributes most to the winter OA (Jiang et al., 2019b), and therefore higher effects from vapor wall loss correction on SOA are observed compared to other sites. At SPC, the fog scavenging processes played an important role on OA during the measurements (Gilardoni et al., 2014), however, the meteorological model failed to reproduce the fog events due to the coarse resolution in this study (Jiang et al., 2019b). Consequently, both VBS_WLS and noWLS lead to an overestimation of OA and SOA, while SOAP and VBS_BASE show better performance probably due to compensation of errors (Fig. S4e). In Bologna, a significant overestimation of temperature was found on 2 to 6 December (Jiang et al., 2019b), leading to a significant underestimation of SOA for all the OA schemes (Fig. S4d). Excluding this period, the modeled SOA by VBS_WLS is 89% higher than the measurements, while the modeled SOA concentrations by the other schemes are closer to the measurements with relative differences of -64% for SOAP, -10% for VBS_BASE, and 4% for VBS_noWLS.

The distinct performance of vapor wall loss corrected VBS at different sites could arise from various reasons. It might come from the high uncertainties of S/IVOC emissions from biomass burning, which were estimated by the same factor for the whole domain but were reported to have substantial inter-country variations (Denier van der Gon et al., 2015). Missing formation and removal processes such as photolytic and heterogeneous oxidation in the model could also result in different model performance for specific sites. In addition, in spite of the advanced chamber measurements we used to optimize the yield parameters, covering a wide range of precursor species and multiple temperature and chamber conditions, the fitted vapor wall loss corrected parameterization is still highly uncertain. To achieve a more robust parameterization and to further improve the model performance for OA, more studies on S/IVOC emissions, as well as the formation and removal mechanisms of SOA based on extensive laboratory studies and field observations with higher spatial and temporal coverage are needed.

### 4.3 Effects of vapor wall loss correction on modeled OA in Europe

**4.3.1 OA**

The modeled OA results in Europe for the whole year 2011 by different OA schemes were compared to investigate the effects of OA schemes and the vapor wall loss correction. Among all the sources, residential biomass burning contributed to 16.3-52.6% of POA and 5.9-28.9% of SOA in winter (Jiang et al., 2019b), indicating the potential roles of vapor wall loss for the biomass burning sector. Figure 6 shows the modeled OA, SOA and POA in winter (December–January–February). VBS_WLS

leads to the highest domain average OA (2.3 µg m$^{-3}$), which is 16.4%, 26.2%, 38.7% and 106.6% higher than VBS_3POA, VBS_noWLS, SOAP and VBS_BASE, respectively. The VBS schemes generally produce higher OA than SOAP, except for the default parameterization (VBS_BASE) in which the lack of SVOC emissions is not considered. However, SOAP leads to the second highest SOA after VBS_WLS, especially in northern Europe where the monoterpene emissions from coniferous forests are relatively high. This is mostly because of the high terpene SOA yields in SOAP2.1, which were reduced in the later

version of the CAMx model (CAMx v7.0, http://www.camx.com). The vapor wall loss corrected yields lead to increased SOA in large areas of central and southern Europe (Fig. 7). The largest difference is predicted for the Po Valley and Romania regions with high residential biomass burning impact (Fig. S5). The overall relative differences between VBS_WLS and VBS_noWLS are more than 80% and the highest grid-scale increment reaches 5.6 µg m$^{-3}$ in the region of Balkans. The modeled POA concentrations are similar to those in the VBS case with correction for SVOC (3POA, noWLS, WLS) with domain average

concentrations ranging from 0.9 (noWLS) to 1.1 (3POA) µg m$^{-3}$, and therefore no significant effects were observed from vapor wall loss correction (Fig. 6). The POA simulated by VBS_BASE (0.3 µg m$^{-3}$) is even lower than SOAP (0.7 µg m$^{-3}$), as the POA is semi-volatile and could evaporate and react with oxidants to form secondary products in VBS while SOAP assumes the POA to be inert.

The effects of different VBS schemes on OA are much smaller in summer (Fig. S6). Despite a slight increase from the

340 VBS_BASE (1.2 µg m$^{-3}$), the modeled OA by the three modified VBS schemes are quite similar (1.4 – 1.5 µg m$^{-3}$). The effects of vapor wall loss corrected yields for biomass burning emissions are negligible due to low emissions in summer (Fig. S7). SOAP produced the highest OA (2.1 µg m$^{-3}$) in summer due to the high SOA yields from monoterpenes as explained before.

### 4.3.2 Fraction of SOA in OA

The effects of the updated VBS schemes on the fraction of annual average SOA in total OA (fSOA = SOA/OA) are shown in

Fig. 8. The VBS schemes lead to a higher fSOA (domain average 71.4%–87.3%) compared to SOAP (domain average 69.9%) in most of the domain except for northern Europe, where SOAP produces high biogenic SOA. The increased POA emissions to offset the missing SVOC emissions (3POA, noWLS, WLS) decrease the fSOA compared to the default VBS parameterization (BASE), while the vapor wall loss correction yields (WLS) result in ~5.8% higher fSOA than noWLS for the domain average and the largest grid-scale increase reaches 43.4% in the Balkans. The absolute differences between fSOA for

WLS and noWLS are relatively higher in rural areas than urban areas, where fSOA is lower due to high primary emissions.

The modeled fSOA values were compared with the measurements from previous studies in Europe (Crippa et al., 2014; Jiang et al., 2019b). The measured fSOA from literature covered 18 sites and different seasons between 2008 and 2011 (Table S2). SOAP tends to underestimate the fSOA, while VBS_BASE significantly over-predicts the fSOA (Fig. 9). Both WLS and noWLS tend to underestimate the high fSOA and overestimate the low fSOA. VBS_WLS has 5% higher fSOA than VBS_noWLS and shows the highest agreement on the range of fSOA with the measurements, as well as the average fSOA values (measured: 69.6%; VBS_WLS: 69.1%). The largest improvements occur in winter, when the vapor wall corrected yields of biomass burning emissions largely increase the SOA production.

## 5. Conclusions

In this study, we optimized the SOA yields for a VBS-based box model using 14 chamber experiments with biomass burning and implemented the fitted VBS parameters (SOA yields, IVOC emissions from biomass burning, and enthalpy of vaporization) in the regional air quality model CAMx v6.5. The influence of the vapor wall loss correction on the model performance was investigated by comparing modeled primary and secondary OA with the traditional and modified OA schemes, including the 2-product approach (SOAP), the standard VBS (VBS_BASE), VBS with 3 times of POA to compensate for the missing SVOC (VBS_3POA), VBS with vapor wall loss correction (VBS_WLS) and an additional reference scenario with the same parameterizations as in VBS_WLS except for using the default SOA yields from biomass burning IVOC (VBS_noWLS).

The vapor wall loss correction increases the mass distributed in the low-volatility bins ($logC^* \leq 0$) by a factor of 4.3, and increases the SOA yields by a factor of 1.9–4.9 (at 298 K). Comparison of the modeled results with different OA schemes with the field measurements from five ACSM/AMS stations in Europe in winter, suggests that VBS_WLS generally has the best performance to predict OA, which lowers the highest mean fractional bias from -72.9% (VBS_BASE) to -1.6% for OA, and -77.8% (SOAP) to 20.0% for SOA. In Europe, the VBS_WLS produces the highest domain average OA in winter (2.3 µg m$^{-3}$), which is 106.6% and 26.2% higher than VBS_BASE and VBS_noWLS, respectively. The largest influence of vapor wall loss correction was predicted in Romania where the VBS_WLS increase the SOA by ~80% compared to VBS_noWLS due to high emissions from residential biomass burning. VBS_WLS also leads to the highest agreement with measurements for the SOA fraction in OA (fSOA) from literature.

The optimized parameterization with vapor wall loss correction in this study is expected to provide some insight to improve SOA underestimation in CTMs. Despite the overall improvement of model performance for predicting SOA, the VBS_WLS was found to increase the mean bias at specific sites compared to noWLS. To achieve a more robust parameterization and to further improve the model performance, complementary studies on S/IVOC emissions, as well as on the formation and removal mechanisms of SOA based on extensive laboratory studies and field observations with higher spatial and temporal coverage are still needed.

*Code and data availability.* The source code of the standard CAMx model is available at the RAMBOLL website (http://www.camx.com). The modified CAMx codes, as well as the source code of the MATLAB-based VBS box model are available online at https://doi.org/10.5281/zenodo.3998342 (Jiang, 2020). Data of the figures are at https://doi.org/ 10.5281/zenodo.4267890. (Jiang, 2021)

*Author Contribution.* JJ and IEH conceived the study. JJ carried out the model simulation and data analysis. GS and AB conducted the chamber measurements. NM, FC, JEP, OF and SG provided the measurement data. SA, ASHP and UB supervised the entire work development. The manuscript was prepared by JJ. All authors discussed and contributed to the final paper.

*Competing interests.* The authors declare that they have no conflict of interest.

**Acknowledgements**. We would like to thank the support of the SNF project SAOPSOAG (200021_169787). We thank the European Centre for Medium-range Weather Forecasts (ECMWF) for the meteorological data, the National Aeronautics and Space Administration (NASA) and its data-contributing agencies (NCAR, UCAR) for the TOMS and MODIS data, the global air quality model data and the TUV model. We thank the support of CAMx by RAMBOLL. Simulation of WRF and CAMx models were performed at the Swiss National Supercomputing Centre (CSCS). We thank the Aerosol, Clouds and Trace gases Research InfraStructure (ACTRIS) and the Chemical On-Line cOmpoSition and Source Apportionment of fine aerosoL (COLOSSAL) cost action (CA16109) for support and harmonization within OA measurements and data treatments.

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

**Table 1:** Description about the different OA schemes.

| OA scheme | IVOB[a] emissions | $k_{OH}$ for IVOB $(cm^3\ molec^{-1}\ s^{-1})$ | SOA yields for IVOB (ppm/ppm)[b] |
|---|---|---|---|
| SOAP | = 4.5*POA_BB | 1.34 | / [c] |
| VBS_BASE | | 4.0 | [0.081, 0.135, 0.800, 0.604, 0.0] |
| VBS_3POA | | 4.0 | [0.081, 0.135, 0.800, 0.604, 0.0] |
| VBS_noWLS | = 12*POA_BB | 1.5 | [0.014, 0.036, 0.076, 0.136, 0.44] |
| VBS_WLS | | 1.5 | [0.078, 0.118, 0.157, 0.177, 0.312] |

[a]IVOB is the abbreviation of "IVOC from Biomass Burning" in CAMx

[b]The yield values are corresponding to volatility bins with saturation concentrations of $10^{-1}$, $10^0$, $10^1$, $10^2$ and $10^3$ µg m$^{-3}$.

[c]SOAP does not separate IVOC from biomass burning and other anthropogenic sectors, and therefore is not comparable with the SOA yields for IVOBs.

**Table 2:** Statistical results for model performance on simulating OA, SOA and POA. The number of daily average observations from five ACSM/AMS stations is 216.

| Species | OA scheme | MB ($\mu g\ m^{-3}$) | ME ($\mu g\ m^{-3}$) | RMSE ($\mu g\ m^{-3}$) | MFB (%) | MFE (%) | r |
|---|---|---|---|---|---|---|---|
| OA | SOAP | -4.1 | 4.9 | 7.2 | -44.3 | 65.3 | 0.38 |
| | VBS_BASE | -4.9 | 5.6 | 7.9 | -72.9 | 83.3 | 0.29 |
| | VBS_3POA | -1.6 | 4.3 | 6.5 | -12.4 | 51.7 | 0.42 |
| | VBS_noWLS | -1.9 | 4.3 | 6.5 | -17.4 | 52.7 | 0.41 |
| | VBS_WLS | -0.4 | 4.6 | 6.9 | -1.6 | 52.2 | 0.41 |
| SOA | SOAP | -2.3 | 3.1 | 4.3 | -77.8 | 98.3 | 0.12 |
| | VBS_BASE | -1.6 | 2.8 | 4.1 | -63.0 | 90.6 | 0.22 |
| | VBS_3POA | -1.2 | 2.8 | 4.1 | -51.1 | 84.3 | 0.23 |
| | VBS_noWLS | -1.3 | 2.8 | 4.0 | -52.5 | 84.9 | 0.24 |
| | VBS_WLS | 0.2 | 3.2 | 4.6 | -20.0 | 76.4 | 0.26 |
| POA | SOAP | -0.7 | 1.9 | 3.1 | 4.4 | 56.7 | 0.49 |
| | VBS_BASE | -2.3 | 2.5 | 4.0 | -64.1 | 81.5 | 0.44 |
| | VBS_3POA | 0.8 | 2.4 | 3.4 | 36.3 | 64.2 | 0.45 |
| | VBS_noWLS | 0.4 | 2.2 | 3.2 | 30.1 | 61.9 | 0.45 |
| | VBS_WLS | 0.6 | 2.3 | 3.3 | 32.4 | 62.5 | 0.45 |

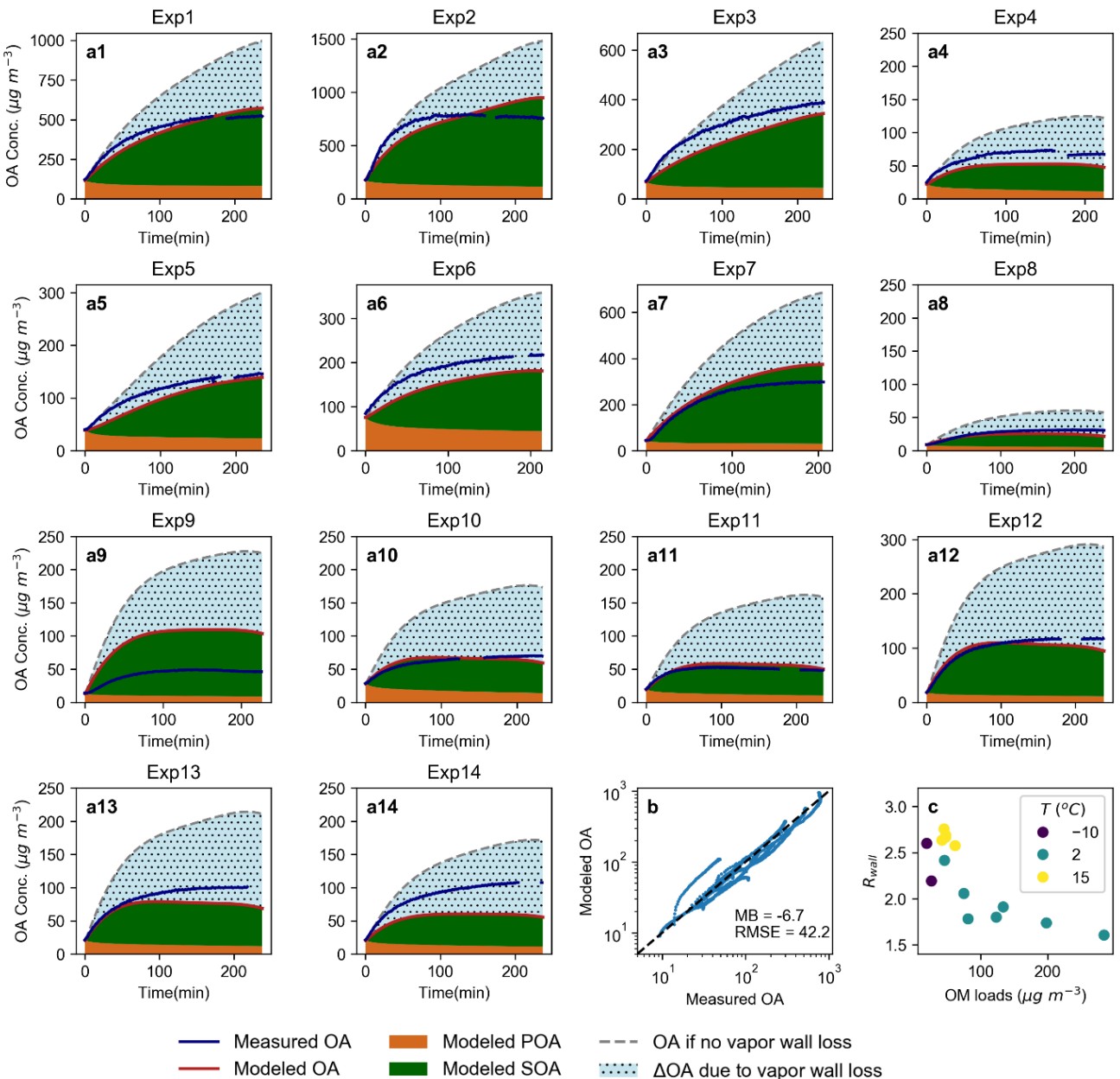

**Figure 1:** Comparison between measured and modeled OA with optimized parameterization under $k_w = 0.0033$ s$^{-1}$, $C_{wall} = 5$ mg m$^{-3}$ (a, b) and relation between the endpoint wall loss factor $R_{wall}$ of each experiment and initial OM loads under different temperature (c). The gray dashed lines in (a) represent modeled OA with the same parameterization but set $k_w=0$.

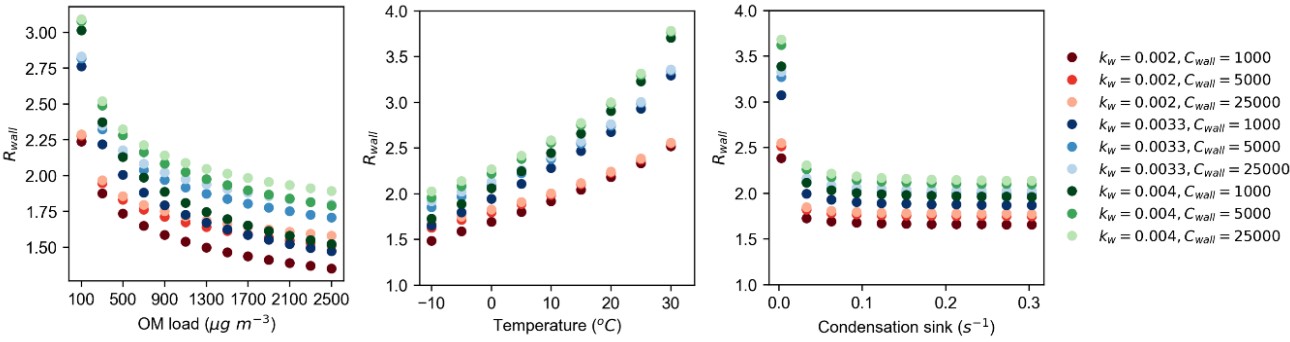

**Figure 2:** Dependence of wall loss factor $R_{wall}$ ($C_{OA,\ kw=0}$ /$C_{OA,\ optimal\ kw}$) on initial organic mass load, temperature and condensation sink.

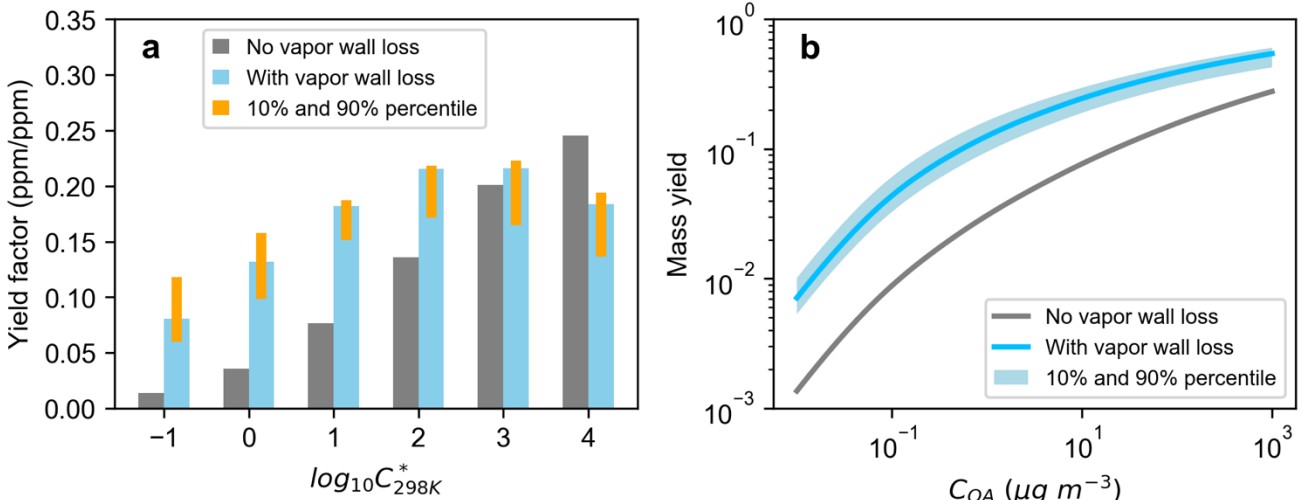

**Figure 3:** Optimized yield factors (a) and mass yield of SOA from biomass burning at 298 K (b) with and without vapor wall loss correction. The blue bars (a) and line (b) with vapor wall loss refer to median chamber conditions with $k_w$ = 0.0033 s$^{-1}$, $C_{wall}$ = 5 mg m$^{-3}$.

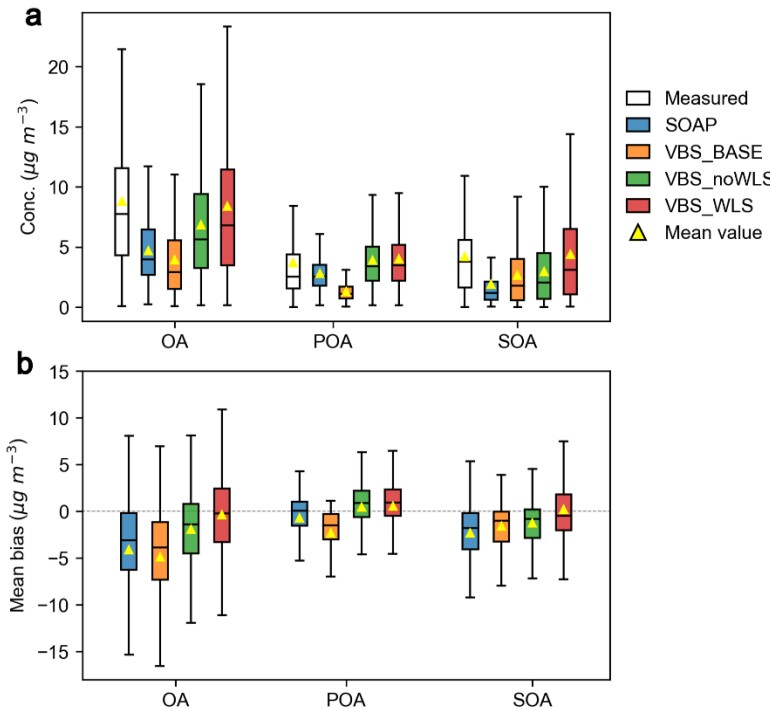

**Figure 4:** Concentrations of measured and modeled OA, POA and SOA at five ACSM/AMS stations in winter (a) and mean bias for different OA schemes (b). The lines inside boxes represent median values, and the yellow triangles represent mean values.

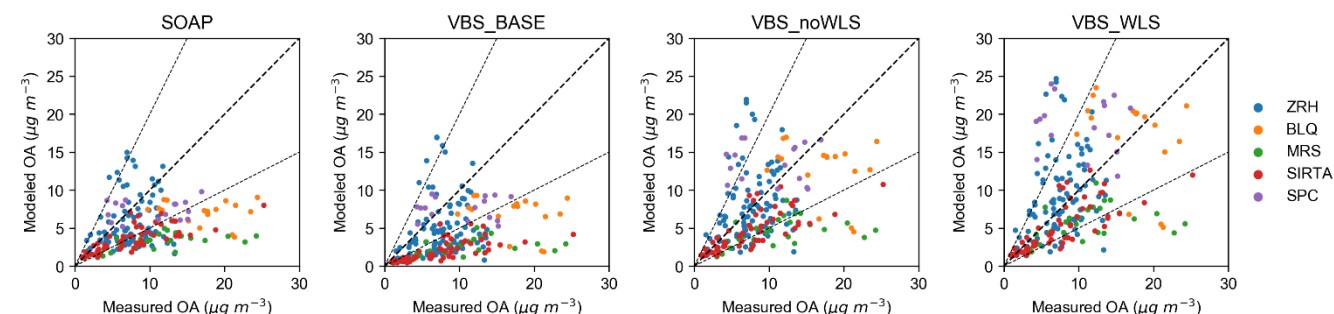

**Figure 5:** Measured and modeled daily average OA using different OA schemes in winter. ZRH: Zurich, BLQ: Bologna, MRS: Marseille, SIRTA: Paris SIRTA, SPC: San Pietro Capofiume.

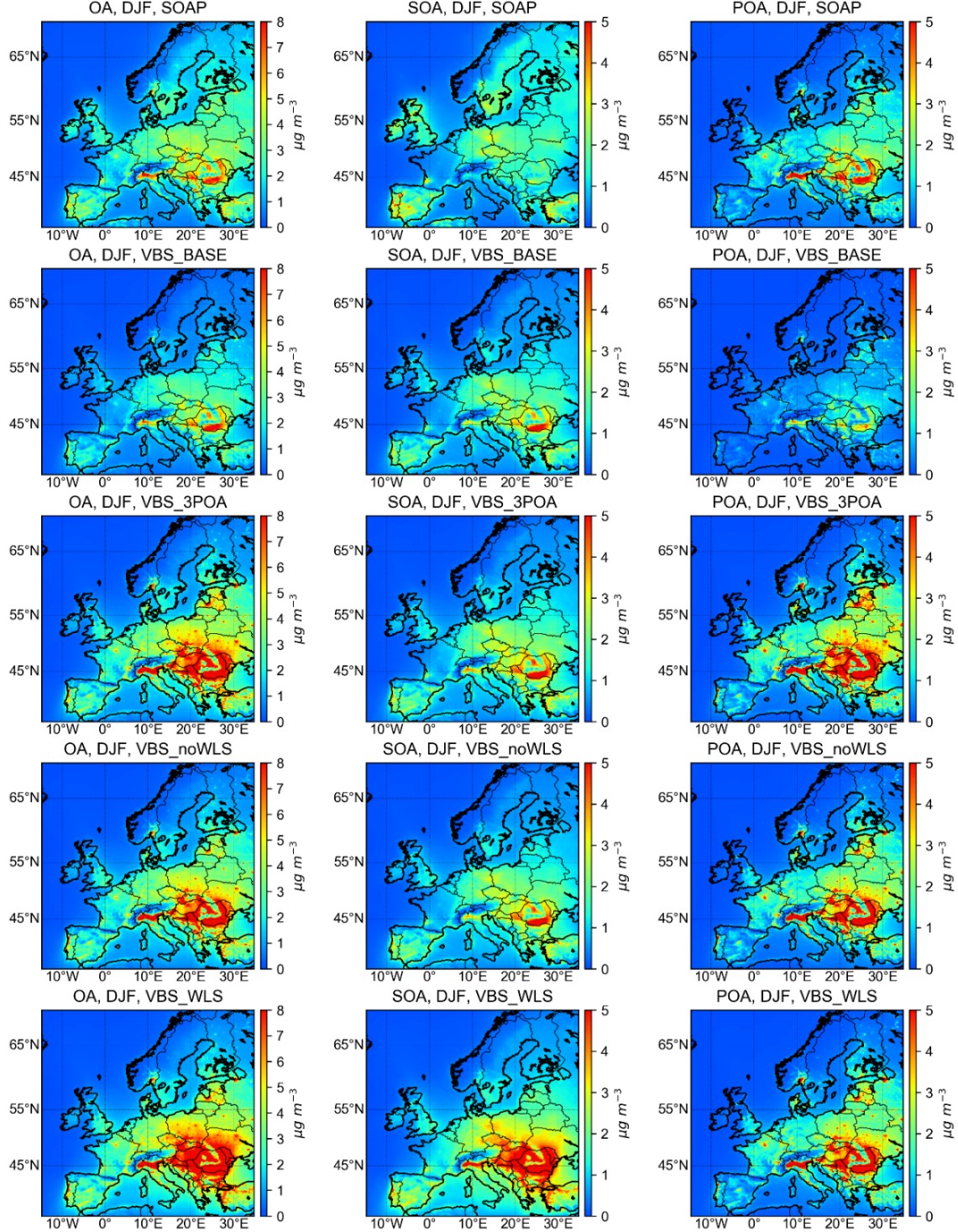

**Figure 6:** Modeled OA, SOA and POA in winter (DJF, December–January–February) by different OA schemes.

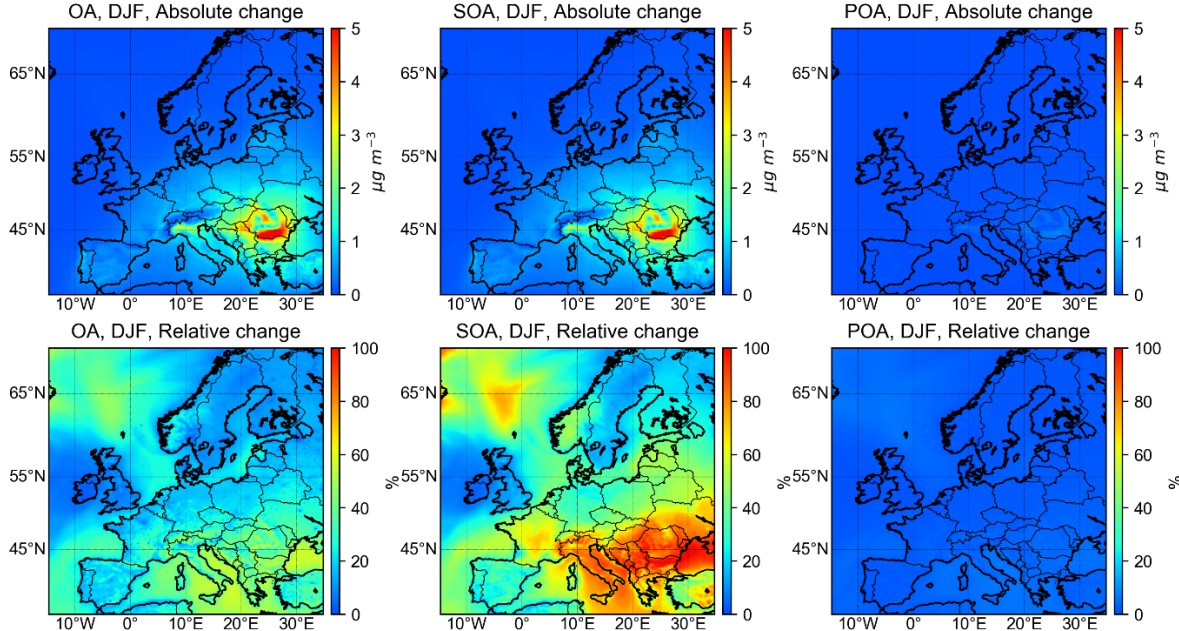

**Figure 7:** Differences in modeled OA, SOA and POA in winter (DJF, December–January–February) by VBS schemes with (VBS_WLS) and without (VBS_noWLS) vapor wall corrections.

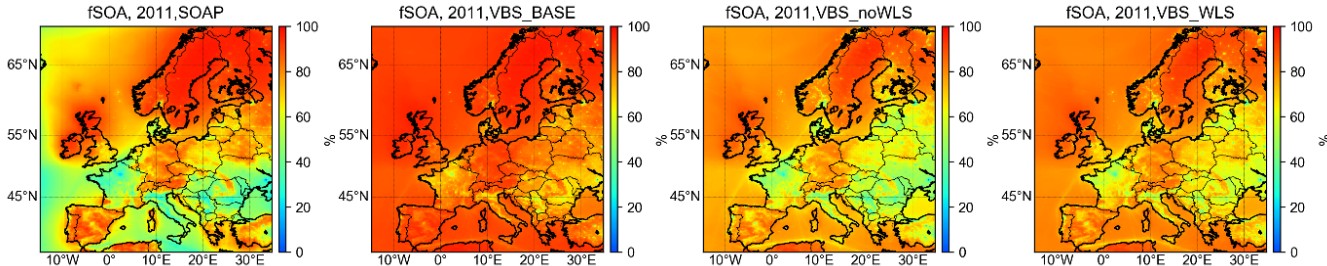

**Figure 8:** Modeled fractions of annual mean SOA to total OA (fSOA) using different OA schemes. Modeled results of VBS_3POA are very similar to VBS_noWLS, and therefore are not shown here.

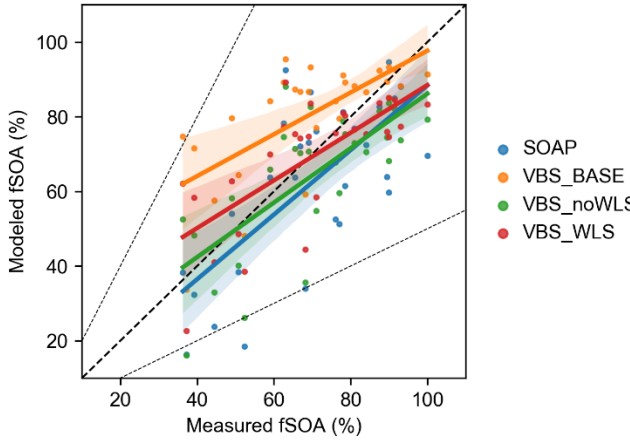

**Figure 9:** Comparison between modeled and measured fSOA from literature over the year (see data and sources in Table S2). The shadows are confidence intervals of the regression lines.