# Peer review of "Influence of biomass burning vapor wall loss correction on modeling organic aerosols in Europe by CAMx v6.50"

_Geoscientific Model Development, 2020_

## Referee Comment (RC1) · Anonymous Referee #1 · 4 Nov 2020

Jiang et al. have performed a modeling study, first to determine parameters to model secondary organic aerosol (SOA) formation from biomass burning and second to use those parameters to study impacts on ambient organic aerosol in Western Europe. They find that accounting for vapor wall losses in chamber experiments seems to generally improve model performance for the OA system (OA, primary OA, and SOA) over the modeling domain, although there are exceptions to the improvement.

This study attempts to better understand the contribution of residential wood combustion to OA concentrations in Western Europe. RWC, referred to as biomass burning in this study (if this is true, this needs to be explicitly stated in the paper to avoid this

source being mistaken for wildfires), is a major source of air pollution in this part of the world and hence the study is well motivated. The chamber modeling and atmospheric modeling approach is justified although a lot of details are missing. The interpretation largely follows from the model results but could benefit from a more detailed description. I am inclined to recommend publication to Geoscientific Model Development after the authors have had a chance to respond to my comments.

Major comments:

1. SOA precursors: I am familiar with this group's earlier work (i.e., Bruns, Stefenelli) where the SOA formation was modeled by knowing the detailed gas-phase precursors and their corresponding SOA mass yields. However, in this work, it is not at all clear what the SOA precursors are and how they are dealt with, with respect to their emissions, oxidation chemistry, and SOA parameters. For instance, what are the SOA precursors? How are their emissions determined? What are the oxidation rate constants? Is there any accounting done for precursor composition? Are all SOA precursors lumped into a single model precursor? Or are all species modeled separately with their own oxidation rate but with the same volatility distribution for their oxidation products? Were different functional forms of the volatility distribution explored and what was the rationale to pick a normal distribution? Answers to these questions and more need to be described in much more detail in the methods section for the reader to follow the general approach. A further point of confusion is how this is then translated to be used in CAMx in Section 3.2

2. OA schemes: I was a little confused about the diversity of schemes tested in this work, particularly if the goal was to examine the specific influence of vapor wall losses on ambient OA concentrations. In other words, shouldn't the work have focused only on VBS_WLS and VBS_noWLS? At present the other simulation results are preventing a clear interpretation of the model results. Furthermore, I was concerned about the treatment of SOA precursors in the two VBS parameterizations: VBS_BASE and VBS_3POA. There is emerging evidence, including that from previous publications

from this group, that SOA precursors from biomass burning are poorly represented in atmospheric models. Along with that evidence though, are detailed observations of what those SOA precursors are, as SVOCs, IVOCs, and VOCs; see for example Hatch et al. (ACP, 2015), Hatch et al. (ACP, 2017), Jen et al. (2019), Koss et al. (ACP, 2018), Sekimoto et al., (ACP, 2018). These studies provide sufficient data to inform the treatment of SVOC, IVOC, and VOC emissions in this work and hence I am concerned that the treatment in this work is a little dated. The current approach to determine IVOC emissions by multiplying POA by 4.5 and SVOC emissions by multiplying POA by 3 seems a bit too much. These S/IVOC estimates need to be reconciled with the extensive literature that I have alluded to earlier. Additionally, I am unclear how the VOCs are dealt with in CAMx. Are these mapped to the traditional SOA precursor species (e.g., benzene, toluene, etc.)? Finally, are vapor wall loss effects accounted for, for the non-biomass burning sources and SOA precursors? What approach is used to do this?

Minor comments:

1. Lines 37-41: Citations are a little dated, replace with more recent studies/reviews.

2. Line 52: Akherati et al. (ES&T, 2020) recently looked at the impacts of vapor wall losses on SOA production from biomass burning emissions. Consider citing.

3. Line 53: Zhang et al. (PNAS, 2014)'s conclusions are sensitive to NOx where the enhancement is much lower at high NOx conditions. Consider stating this.

4. Line 59-60: Not sure I agree with this statement of an enhancement larger than 4.

5. Line 76: Remove 'datasets from'.

6. Line 86: Unclear what 'be diluted' means. The experiments need to be described in more detail in Section 2.1 for the reader to connect the modeling approach to the how the experiments were conducted.

7. Lines 87-92: What is the rationale for not accounting for losses of condensable

vapors to the particles on the walls?

8. Line 97: Both Cappa et al. (ACPD, 2020) and Akherati et al. (ES&T, 2020), in chamber experiments performed on biomass burning emissions, found little evidence for POA evaporation with dilution suggesting that the POA was either composed of low-volatility material or there were limitations to evaporation linked to the phase state. What are the implications of the findings from these two studies here?

9. Line 100: It would be useful to compare the vapor wall loss rates used in this work (derived from Bertrand) to similar estimates now made for other chambers: Zhang et al. (PNAS, 2014), Krechmer et al. (ES&T, 2016), Nah et al. (ACP, 2016), He et al. (ESPI, 2020), and Akherati et al. (ES&T, 2020). There might also be a few additional studies from Carnegie Mellon that you could look into. How does that comparison look?

10. Equation 2: Why is particle wall loss not included in here? If the measurement data are already corrected for particle losses, what estimates were used, i.e., w=0, w=1, or average?

11. Line 118: Krechmer et al. (ES&T, 2016) have argued that C_wall needs to be modeled as a function of C* based on observations of loss rates of oxidation products with different C*. Can this also be modeled in this work?

12. Section 3.2: Perhaps, focus this section on the VBS_noWLS and VBS_WLS and frame the other schemes as sensitivity or legacy simulations?

13. Line 162: By how much was the aerosol yield scaled by to account for vapor wall loss?

14. VBS_WLS: How does one think about the relatively higher NOx conditions in the chamber experiments and the much lower NOx conditions in the atmosphere while translating the SOA parameters from the chamber to the atmosphere? How can this influence be further studied via sensitivity simulations?

15. Lines 202-203: What was the OM loading in the other experiments that is compared to here?

16. Line 215: Is the temperature effect on gas-phase chemistry taken into account?

17. Section 4.1, last paragraph: I think I understand what you mean but try to explain the 'vapor wall loss yield' results better since one would expect the yields to be lower when accounting for vapor loss to the walls.

18. Figure 3(a): Are those emission factors or production factors for SOA?

19. Figure 5: What is the $R^2$? Can you describe the model skill and how it compares across schemes, cities, and other comparable models?

20. Figure 6: It is surprising to see so much SOA production compared to POA even in the winter. Are aqueous pathways for SOA production accounted for? Were the chamber experiments done under 'dry' or 'wet' conditions?

21. Figure 9: Is this just for winter or all year round?

22. Line 283: Can you make emissions maps to make this point about the dominance of biomass burning emissions in certain locations clear?

23. Section 4: The paper would benefit from doing a sensitivity simulation where biomass burning is eliminated from the emissions to understand the relative contribution of this source to total OA/POA/SOA.

---

## Referee Comment (RC2) · Anonymous Referee #2 · 17 Nov 2020

General Comments

Jiang et al. report a modeling study that: (1) evaluates the impact of vapor wall losses during chamber studies on parameters for SOA formation from residential biomass burning emissions, and (2) simulates the increases in OA and SOA concentrations once a CTM is updated with new SOA parameters that take into account these wall losses. Overall, the manuscript is well-written and addresses an important topic in the field of atmospheric aerosol modeling, since implementing accurate parameters for SOA formation in CTMs is challenging. I support publication in GMD once my comments below have been taken into consideration.

[Figure]

Line-by-line comments

Page 1, Line 27 – 29: What is the difference between the "standard VBS" and the "reference scenario"? Does the reference scenario refer to the traditional two-product approach? Please clarify as this is an interesting finding. Less importantly, I also don't understand why the authors have specifically highlighted the result from Romania.

Page 2, Line 37 – 39: The authors imply that residential biomass burning emission is "the dominant source for... secondary organic aerosols in winter". However, this statement is supported exclusively by three European studies that are referenced on line 39. Therefore, the authors need to clarify that this conclusion is specific to the European domain.

Page 2, Line 47: I would kindly suggest that the authors specify that the POA emissions are treated as semi-volatile when this "scaling-up" is performed, since some models still assume that POA is nonvolatile.

Page 2, Line 57: The work of Hayes et al. 2015 concerning vapor wall losses used a box model and not a CTM.

Section 2.1: How does the utilization of beech wood as the only fuel potentially bias the results? Is this a fuel commonly used in residential biomass burning in Europe? Basing the parameterization of the VBS scheme on a single fuel is not necessarily a flaw in the study, but some contextualization is needed here to understand how this limitation might influence the model results.

Line 105: The phrase "gas-phase equilibrium concentrations in particle phase" is not coherent. Please clarify. Furthermore, I don't think equation (3) can be correct. A partitioning coefficient of 1 would indicate complete partitioning to the particle phase, but this would give a $C_{eq}(i,p)$ value of zero. This comment also applies to equation (4).

Line 201: The reasoning why the OM loading would have an effect on the box model's accuracy is not clear. Please elaborate. Also, there are some runs when the loadings

are low when the model accuracy is very reasonable, for example a11, so the OM loading does not seem to explain by itself the poor accuracy of the model observed for experiments 9 and 14.

Figure 1: It would be useful if a table of the experiment conditions was provided.

Line 203 – 204: I think the text contains an error here. If anything there is an underestimation at short times and an overestimation at long times. More generally, it seems like the comparison between the model and the measurement varies a lot between experiments, so it is difficult to make conclusions regarding whether the box model is overestimating or underestimating.

Lines 205 – 207: I think using percentages here to compare the two box model versions (with or without vapor wall losses) overstates the difference between the models. In the end, the differences in the MB and RMSE are only about 6 ug/m3, which is not very much when most of the experiments are run at OM concentrations near or above 100 ug/m3.

Figure 3: I very strongly suggest that these data also be given in a table so that the quantitative results can be used by other researchers.

Lines 284 – 287: This sentence is confusing. Which model cases are specifically being compared? In addition, in Figure 7, only the schemes VBS_WLS and VBS_noWLS are compared, but then in the text the 3POA scheme is mentioned as well.

Figure 8: It would be helpful to specify in the figure caption that these plots are annual averages. In addition, why are annual averages used and discussed rather than wintertime measurements, as is done in the other sections of the manuscript?

Lines 318 – 322: The comparisons summarized here are rather haphazard. First, for OA, the VBS_WLS scheme is compared to the VBS_BASE scheme. Then next, for SOA, the VBS_WLS scheme is compared to the SOAP scheme. The authors should be consistent in what schemes they are comparing to as "base cases".

---

## Author Comment (AC1) · 15 Dec 2020

**Responses to the comments of anonymous referee #1**

We thank the referee for the valuable comments that have greatly helped us to improve the manuscript. Please find below our responses (in black) after the referee comments (in blue). The changes in the revised manuscript are written in *italic*.

Jiang et al. have performed a modeling study, first to determine parameters to model secondary organic aerosol (SOA) formation from biomass burning and second to use those parameters to study impacts on ambient organic aerosol in Western Europe. They find that accounting for vapor wall losses in chamber experiments seems to generally improve model performance for the OA system (OA, primary OA, and SOA) over the modeling domain, although there are exceptions to the improvement.

This study attempts to better understand the contribution of residential wood combustion to OA concentrations in Western Europe. RWC, referred to as biomass burning in this study (if this is true, this needs to be explicitly stated in the paper to avoid this source being mistaken for wildfires), is a major source of air pollution in this part of the world and hence the study is well motivated. The chamber modeling and atmospheric modeling approach is justified although a lot of details are missing. The interpretation largely follows from the model results but could benefit from a more detailed description. I am inclined to recommend publication to Geoscientific Model Development after the authors have had a chance to respond to my comments.

We added a sentence in the end of the Introduction (P3, L76–77) to clarify that the biomass burning in this study refers to residential biomass burning.

"The biomass burning in this study refers to residential biomass burning, while the wildfires and prescribed burning are not included."

**Major comments:**

1. SOA precursors: I am familiar with this group's earlier work (i.e., Bruns, Stefenelli) where the SOA formation was modeled by knowing the detailed gas-phase precursors and their corresponding SOA mass yields. However, in this work, it is not at all clear what the SOA precursors are and how they are dealt with, with respect to their emissions, oxidation chemistry, and SOA parameters. Our study is based exactly on the same experimental datasets of Stefenelli et al. (2019) and Bruns et al. (2016). Many details about the datasets we used, such as the SOA precursors, have already been described in the previous studies the referee mentioned (Bruns et al, 2017; Stefenelli et al., 2019), and therefore were not included in the manuscript.

**What are the SOA precursors? How are their emissions determined? Is there any accounting done for precursor composition?**

A common set of 263 ions was extracted from the PTR-ToF-MS, and among these 86 were identified as potential SOA precursors, which are listed in Table S1 of Stefenelli et al. (2019). We determine the emissions from the observed compound decay rates measured by the PTR, and the production rates of condensable species taking into account their loss by dilution.

Are all SOA precursors lumped into a single model precursor? Or are all species modeled separately with their own oxidation rate but with the same volatility distribution for their oxidation products? Yes, all the precursors are lumped into one category, and are assumed to have the same reaction rate and volatility distribution of their oxidation products.

What are the oxidation rate constants?

The oxidation rate constant for the lumped precursor towards OH used in the model is calculated based on the measurements, and the value is  $1.5 \times 10^{-11}$  cm3 molec-1 s-1. It is shown in the new Table 1 in the revised manuscript and the source code we uploaded.

**Were different functional forms of the volatility distribution explored and what was the rationale to pick a normal distribution?**

In Stefenelli et al (2019) we have explored other functional forms, but the resulting yields were not sensitive to the assumed function. Using the normal distribution ensures positive  $\zeta_i$  values, allows constraining of the total mass fraction of the certain surrogate equals 1, and reduce the model's degree of freedom significantly. Therefore, we picked the normal distribution in this study. An explanation is added in P\*L\*.

Answers to these questions and more need to be described in much more detail in the methods section for the reader to follow the general approach.

We have updated the methodology section as the referee suggested by adding more details related to the modelling part:

"In the model, we assumed that the condensable gases generated from the oxidation of precursors could 1) partition to the particle phase, 2) be lost on the chamber wall, and 3) be diluted by other gases injected into the smog chamber. To simplify the simulation, as well as to facilitate its implementation in the VBS scheme of the regional model CAMx, all the SOA precursors were lumped into one surrogate with the same reaction rate, determined based on PTR measurements  $(1.5 \times 10^{-11} \text{ cm}^3 \text{ molec}^{-1} \text{ s}^{-1})$ . One volatility distribution was assumed for the oxidation products of the lumped precursors." (P3, L94–P4,L98)

"The production rate of oxidized organic gases (P) is used as inputs for the box model. It is determined by the consumption rates of precursors measured by PTR taking into account their dilution." (P4, L106–L107)

"The assumption of normal distribution could ensure positive  $\zeta_i$  values, allow constraining the total mass fraction of the certain surrogate equals 1, and reduce the model's degree of freedom significantly, as reported in Stefenelli et al. (2019)" (P4, L112–L114)

A further point of confusion is how this is then translated to be used in CAMx in Section 3.2.

The optimized parameters can be directly used in CAMx by simply changing the parameter values of the source code. We added a sentence in P7, L200–L202 to guide the readers to the modified source code.

"- VBS\_WLS. ... The modified parameters for volatility bin specific yields and  $\Delta H_{vap}$  of the oxidized products from IVOC can be found at https://doi.org/10.5281/zenodo.3998342"

2. OA schemes: I was a little confused about the diversity of schemes tested in this work, particularly if the goal was to examine the specific influence of vapor wall losses on ambient OA concentrations. In other words, shouldn't the work have focused only on VBS\_WLS and VBS\_noWLS? At present the other simulation results are preventing a clear interpretation of the model results.

It is true that the focus of this study is the influence of vapor wall loss correction. However, we wanted to put into perspective the resulting improvement from the wall loss correction compared to results from other modifications/parameterizations that are currently strongly debated in the community. Therefore, we included SOAP and VBS\_BASE which represent the two standard parameterizations in CAMx. VBS\_3POA was added as it represents a common approach to offset the missing SVOC emissions in recent modelling studies without vapor wall loss. VBS\_noWLS is

another reference case for that without vapor wall loss, which uses exactly the same parameters as VBS\_WLS except for the SOA yields from IVOCs. Both VBS\_3POA (based on default parameterization of CAMx) and VBS\_noWLS (optimized in this study assuming  $k_w=0$ ) assume that there is no vapor wall loss, and are supposed to have similar results. To further clarify the necessity of all the scenarios, we added a paragraph in P6, L173–L179 and a new Table 1 to introduce the purpose of each scheme.

"To investigate the effects of vapor wall loss corrected yields, as well as to compare to other modifications/parameterizations that are currently strongly debated in the community, five simulations with different OA schemes were conducted in this study (Table 1). Besides VBS\_WLS which uses the optimized parameterization with vapor wall loss correction for the biomass burning sector, SOAP and VBS\_BASE represent the two standard parameterization in CAMx; VBS\_3POA represents a common approach to offset the missing SVOC emissions in recent modelling studies without vapor wall loss; VBS\_noWLS is another reference case for that without vapor wall loss, which uses exactly the same parameters as VBS\_WLS except for the SOA yields from IVOCs. Details about each OA schemes are introduced below:..."

| OA scheme | IVOB a emissions | k OH for IVOB                          | SOA yields for IVOB (ppm/ppm) b |
|-----------|-----------------------------|---------------------------------------------------|--------------------------------------------|
|           |                             | $(\text{cm}^3 \text{ molec}^{-1} \text{ s}^{-1})$ |                                            |
| SOAP      | $= 4.5 * POA_BB$            | 1.34                                              | /°                                         |
| VBS_BASE  |                             | 4.0                                               | [0.081, 0.135, 0.800, 0.604, 0.0]          |
| VBS_3POA  |                             | 4.0                                               | [0.081, 0.135, 0.800, 0.604, 0.0]          |
| VBS_noWLS | = 12*POA_BB                 | 1.5                                               | [0.014, 0.036, 0.076, 0.136, 0.44]         |
| VBS_WLS   |                             | 1.5                                               | [0.078, 0.118, 0.157, 0.177, 0.312]        |

Table 1: Description about the different OA schemes.

a IVOB is the abbreviation of "IVOC from Biomass Burning" in CAMx

bThe yield values are corresponding to volatility bins with saturation concentrations of  $10^{-1}$ ,  $10^{0}$ ,  $10^{1}$ ,  $10^{2}$  and  $10^{3} \,\mu g \,m^{-3}$ .

cSOAP does not separate IVOC from biomass burning and other anthropogenic sectors, and therefore is not comparable with the SOA yields for IVOBs.

Furthermore, I was concerned about the treatment of SOA precursors in the two VBS parameterizations: VBS\_BASE and VBS\_3POA. There is emerging evidence, including that from previous publications from this group, that SOA precursors from biomass burning are poorly represented in atmospheric models. Along with that evidence though, are detailed observations of what those SOA precursors are, as SVOCs, IVOCs, and VOCs; see for example Hatch et al. (ACP, 2015), Hatch et al. (ACP, 2017), Jen et al. (2019), Koss et al. (ACP, 2018), Sekimoto et al., (ACP, 2018). These studies provide sufficient data to inform the treatment of SVOC, IVOC, and VOC emissions in this work and hence I am concerned that the treatment in this work is a little dated. The current approach to determine IVOC emissions by multiplying POA by 4.5 and SVOC emissions by multiplying POA by 3 seems a bit too much. These S/IVOC estimates need to be reconciled with the extensive literature that I have alluded to earlier.

We totally agree with the referee that recent publications have provided considerable information to improve the estimates of S/IVOC emissions. In VBS\_noWLS, we determined the IVOC emissions from the experimental data that the IVOC/POA(SVOC) to be ~13.7 (Fig. S1). This ratio was relatively similar between different burns. For the primary SVOCs, we have treated them with a more routine approach (POA+SVOC=3\*POA) used in previous studies (Tsimpidi et al., 2010; Shrivastava et al, 2011; Ciarelli et al., 2016; Ciarelli et al., 2017). This allows focusing on the effect of wall losses

of IVOC oxidation products on modelling secondary BBOA and a direct comparison between cases where these losses are taken or not into consideration. We added the recent advances on quantifying S/IVOC emissions in the Introduction section and explained why current approach is used in this study in section 3.2.

"Increasing number of laboratory experimental studies attempted to identify and quantify the S/IVOC precursors from biomass burning, and found it is of high variability depending on different burning conditions (Hatch et al., 2015, Hatch et al., 2017; Jen et al., 2019; Koss et al., 2018; Sekimoto et al., 2018). In order to compensate the effects from missing precursors, various modeling studies treated the POA as semi-volatile and increased the POA emissions by a factor based on findings from chamber experiments." (P2, L46–L48)

"- VBS\_3POA. An increasing number of experimental and modeling studies have reported a considerable contribution of semi-volatile organic compounds (SVOCs) to SOA formation (Bruns et al., 2016; Ciarelli et al., 2017b; Denier van der Gon et al., 2015; Hatch et al., 2017; Woody et al., 2015), while the SVOC are absent in the current emission inventories. Despite considerable variabilities of the SVOC emissions from biomass burning according to recent studies, the VBS\_3POA is supposed to be a reference case representing the commonly used approach without vapor wall loss, and therefore we adopted the routine approach – increase the POA emissions by a factor of 3 to offset the influence of missing SVOC emissions. This approach has been widely used in modeling studies (Ciarelli et al., 2016; Ciarelli et al., 2017a; Shrivastava et al., 2011; Tsimpidi et al., 2010). All the other parameters were kept the same as the standard VBS parameterization in CAMx v6.50." (P7, L190–L198)

**Additionally, I am unclear how the VOCs are dealt with in CAMx. Are these mapped to the traditional SOA precursor species (e.g., benzene, toluene, etc.)?**

CAMx has benzene, toluene, xylene and IVOC as the traditional SOA precursor species from anthropogenic emissions. In this study, the optimized parameters were implemented in the set of SOA from biomass burning IVOC. According to our measurements, the IVOC emissions from residential biomass burning is ~13.7 times the primary OM loads (as mentioned in current P7, L203), and the fraction of benzene, toluene and xylene is roughly 15% of the total IVOC emissions. Therefore, we applied a factor of 12 to calculate the IVOC emissions from biomass burning to avoid double counting of benzene, toluene and xylene, which are already included in the emission inputs. We updated P7, L203–L206 to make it clear.

"Based on the chamber measurements, the IVOC emissions from residential biomass burning is ~13.7 times the primary OM load (Fig. S1), among which the traditional precursors toluene, xylene and benzene occupying ~15% of the total emission. To avoid double counting of these traditional precursors which are already included in the emission inventory, we applied a factor of 12 to calculate the IVOC emissions from biomass burning."

**Finally, are vapor wall loss effects accounted for, for the non-biomass burning sources and SOA precursors? What approach is used to do this?**

No, we did not account for the vapor wall loss for other sources. The main focus of this study is the vapor wall loss correction for precursors from *biomass burning* to show the potential importance of correcting for vapor wall losses for complex emissions.

**Minor comments:**

1. Lines 37-41: Citations are a little dated, replace with more recent studies/reviews.

We updated the citations related with the underestimation of biomass burning in CTM in P2, L40–L41.

"Despite its substantial contribution to OA, biomass burning OA is largely underestimated by chemical transport models (CTM) (Ciarelli et al., 2017a; Hallquist et al., 2009; Robinson et al., 2007; Theodoritsi and Pandis, 2019; Woody et al., 2016)."

2. Line 52: Akherati et al. (ES&T, 2020) recently looked at the impacts of vapor wall losses on SOA production from biomass burning emissions. Consider citing.We added Akherati et al. (2020) in P2, L52 as suggested.

3. Line 53: Zhang et al. (PNAS, 2014)'s conclusions are sensitive to NOx where the enhancement is much lower at high NOx conditions. Consider stating this.

We rephrased the sentence to address the influence of  $NO_x$  conditions in P2, L56–L57.

"Zhang et al. (2014) reported that the vapor wall losses may lead to an underestimation of SOA by a factor of 1.1-4.2, depending on different  $NO_x$  conditions"

4. Line 59-60: Not sure I agree with this statement of an enhancement larger than 4. We modified the statement in P2, L62–64 to be more specific.

"Nevertheless, recent studies showed that the vapor wall losses lead to even larger variability on SOA yields according to different chamber conditions and precursor species (Cappa et al., 2016; Akherati et al., 2019)"

5. Line 76: Remove 'datasets from'. Done

6. Line 86: Unclear what 'be diluted' means. The experiments need to be described in more detail in Section 2.1 for the reader to connect the modeling approach to the how the experiments were conducted.

The "diluted" refers to dilution in the smog chamber due to injection of other gases. We rephrased the sentence in L95–96 to make it clear. The details of experiments were actually already introduced in our previous studies (Stefenelli et al., 2019; Bertrand et al., 2017; Bruns et al., 2016). To further clarify it, we added a Table S1 to introduce the conditions of each experiment.

"We assumed that the condensable gases generated from oxidation of the precursors could 1) partition to the particle phase, 2) be lost on the chamber wall, as well as 3) be diluted by other gases injected into the smog chamber" (P3, L95–L96)

"The conditions of each chamber experiment are shown in Table S1. More detailed description of the experiments can be found in Stefenelli et al. (2019), Bertrand et al. (2017) and Bruns et al. (2016)." (P3, L90–L91)

| #Exp | References             | Date       | Experimental temperature (°C) | Stove type a | OM loads
(µg m -3 ) |
|------|------------------------|------------|-------------------------------|-------------------------|-----------------------------------|
| 1    | Bertrand et al. (2017) | 29.10.2015 | 2                             | stove 1                 | 198                               |
| 2    | Bertrand et al. (2017) | 30.10.2015 | 2                             | stove 1                 | 285                               |
| 3    | Bertrand et al. (2017) | 04.11.2015 | 2                             | stove 1                 | 123                               |
| 4    | Bertrand et al. (2017) | 05.11.2015 | 2                             | stove 1                 | 46                                |

Table S1: Experimental conditions for the 14 chamber experiments used in this study.

| Bertrand et al. (2017) | 06.11.2015                                                                                                                                                                                                           | 2                                                                                                                                                                                                                                                                                                           | stove 2                                                                                                                                                                                                                                                                                                                        | 75                                                                                                                                                                                                                                                                                                                                                                                                                                         |
|------------------------|----------------------------------------------------------------------------------------------------------------------------------------------------------------------------------------------------------------------|-------------------------------------------------------------------------------------------------------------------------------------------------------------------------------------------------------------------------------------------------------------------------------------------------------------|--------------------------------------------------------------------------------------------------------------------------------------------------------------------------------------------------------------------------------------------------------------------------------------------------------------------------------|--------------------------------------------------------------------------------------------------------------------------------------------------------------------------------------------------------------------------------------------------------------------------------------------------------------------------------------------------------------------------------------------------------------------------------------------|
| Bertrand et al. (2017) | 07.11.2015                                                                                                                                                                                                           | 2                                                                                                                                                                                                                                                                                                           | stove 2                                                                                                                                                                                                                                                                                                                        | 134                                                                                                                                                                                                                                                                                                                                                                                                                                        |
| Bertrand et al. (2017) | 09.11.2015                                                                                                                                                                                                           | 2                                                                                                                                                                                                                                                                                                           | stove 2                                                                                                                                                                                                                                                                                                                        | 81                                                                                                                                                                                                                                                                                                                                                                                                                                         |
| Bruns et al. (2016)    | 02.04.2014                                                                                                                                                                                                           | -10                                                                                                                                                                                                                                                                                                         | stove 3                                                                                                                                                                                                                                                                                                                        | 19                                                                                                                                                                                                                                                                                                                                                                                                                                         |
| Bruns et al. (2016)    | 17.03.2014                                                                                                                                                                                                           | -10                                                                                                                                                                                                                                                                                                         | stove 3                                                                                                                                                                                                                                                                                                                        | 26                                                                                                                                                                                                                                                                                                                                                                                                                                         |
| Bruns et al. (2016)    | 25.03.2014                                                                                                                                                                                                           | 15                                                                                                                                                                                                                                                                                                          | stove 3                                                                                                                                                                                                                                                                                                                        | 62                                                                                                                                                                                                                                                                                                                                                                                                                                         |
| Bruns et al. (2016)    | 27.03.2014                                                                                                                                                                                                           | 15                                                                                                                                                                                                                                                                                                          | stove 3                                                                                                                                                                                                                                                                                                                        | 45                                                                                                                                                                                                                                                                                                                                                                                                                                         |
| Bruns et al. (2016)    | 28.03.2014                                                                                                                                                                                                           | 15                                                                                                                                                                                                                                                                                                          | stove 3                                                                                                                                                                                                                                                                                                                        | 42                                                                                                                                                                                                                                                                                                                                                                                                                                         |
| Bruns et al. (2016)    | 29.03.2014                                                                                                                                                                                                           | 15                                                                                                                                                                                                                                                                                                          | stove 3                                                                                                                                                                                                                                                                                                                        | 48                                                                                                                                                                                                                                                                                                                                                                                                                                         |
| Bruns et al. (2016)    | 30.03.2014                                                                                                                                                                                                           | 15                                                                                                                                                                                                                                                                                                          | stove 3                                                                                                                                                                                                                                                                                                                        | 48                                                                                                                                                                                                                                                                                                                                                                                                                                         |
|                        | Bertrand et al. (2017)
Bertrand et al. (2017)
Bertrand et al. (2017)
Bruns et al. (2016)
Bruns et al. (2016) | Bertrand et al. (2017)06.11.2015Bertrand et al. (2017)07.11.2015Bertrand et al. (2017)09.11.2015Bruns et al. (2016)02.04.2014Bruns et al. (2016)17.03.2014Bruns et al. (2016)25.03.2014Bruns et al. (2016)27.03.2014Bruns et al. (2016)28.03.2014Bruns et al. (2016)29.03.2014Bruns et al. (2016)30.03.2014 | Bertrand et al. (2017)06.11.20152Bertrand et al. (2017)07.11.20152Bertrand et al. (2017)09.11.20152Bruns et al. (2016)02.04.2014-10Bruns et al. (2016)17.03.2014-10Bruns et al. (2016)25.03.201415Bruns et al. (2016)27.03.201415Bruns et al. (2016)28.03.201415Bruns et al. (2016)29.03.201415Bruns et al. (2016)29.03.201415 | Bertrand et al. (2017)06.11.20152stove 2Bertrand et al. (2017)07.11.20152stove 2Bertrand et al. (2017)09.11.20152stove 2Bruns et al. (2016)02.04.2014-10stove 3Bruns et al. (2016)17.03.2014-10stove 3Bruns et al. (2016)25.03.201415stove 3Bruns et al. (2016)27.03.201415stove 3Bruns et al. (2016)28.03.201415stove 3Bruns et al. (2016)29.03.201415stove 3Bruns et al. (2016)29.03.201415stove 3Bruns et al. (2016)29.03.201415stove 3 |

a Stove 1 manufactured before 2002 (Cheminées Gaudin Ecochauff 625), stove 2 fabricated in 2010 (Invicta Remilly) and stove 3 (Avant, 2009, Attika).

**7. Lines 87-92: What is the rationale for not accounting for losses of condensable vapors to the particles on the walls?**

We neglect losses of condensable species onto wall deposited particles similarly to the majority of chamber studies in literature (Saleh et al., 2013). This is based on the assumption that the surface of particles on the walls is smaller than suspended particle surface or the near-surface vapor concentration differs from the bulk vapor concentration due to diffusion limitations at the wall boundary layer. This treatment provides a lower bound of particle wall loss deposition and hence lowest estimates of SOA yields. In our case, because aerosol mass is rapidly produced, both treatments considering vapors are lost or not on wall deposited particles provide similar results. By contrast, when vapors losses onto chamber walls are considered, aerosol yields are found to be significantly higher.

8. Line 97: Both Cappa et al. (ACPD, 2020) and Akherati et al. (ES&T, 2020), in chamber experiments performed on biomass burning emissions, found little evidence for POA evaporation with dilution suggesting that the POA was either composed of low-volatility material or there were limitations to evaporation linked to the phase state. What are the implications of the findings from these two studies here?

We would like to draw the attention of the reviewer to the work of Sinha et al. 2018 (AST), which found little evidence for particle phase diffusion limitations. Therefore, most likely the slow evaporation observed in Cappa et al. (2020) and Akherati et al. (2020) is likely due to the low volatility of POA. We consider in our case POA volatility distributions from May et al. (2013) and our results also show little evaporation of the POA ( $\sim$ 20%).

9. Line 100: It would be useful to compare the vapor wall loss rates used in this work (derived from Bertrand) to similar estimates now made for other chambers: Zhang et al. (PNAS, 2014), Krechmer et al. (ES&T, 2016), Nah et al. (ACP, 2016), He et al. (ESPI, 2020), and Akherati et al. (ES&T, 2020). There might also be a few additional studies from Carnegie Mellon that you could look into. How does that comparison look?

Comparison between the vapor wall loss rates  $(k_w)$  in this study and other existing studies are added in P4, L115–L120.

"The  $k_w$  varies significantly depending on the chamber conditions. Zhang et al. (2014) reported  $k_w$  values of  $2.5 \times 10^{-4} \text{ s}^{-1}$  and  $1 \times 10^{-4} \text{ s}^{-1}$  for toluene and other VOCs respectively, while it is much higher in recent studies such as  $1.2 \times 10^{-3}$  to  $2.4 \times 10^{-3} \text{ s}^{-1}$  in Krechmer et al. (2016),  $1.28 \times 10^{-3} \text{ s}^{-1}$  in Akherati et al. (2020), and  $\sim 1 \times 10^{-3}$  to  $3.3 \times 10^{-3} \text{ s}^{-1}$  in Bertrand et al. (2018). To cover the wide range of vapor wall loss, we tested three  $k_w$  values  $0.0020 \text{ s}^{-1}$ ,  $0.0033 \text{ s}^{-1}$ ,  $0.0040 \text{ s}^{-1}$  based on the condition of our

chamber. A base case was also developed assuming there is no vapor wall loss in the chamber ( $k_w = 0$ )."

10. Equation 2: Why is particle wall loss not included in here? If the measurement data are already corrected for particle losses, what estimates were used, i.e., w=0, w=1, or average? Yes, the particle wall losses are already corrected in the measurement data. We added a sentence in L 89–L90 to clarify it. We have considered particle wall losses using w=0. w=0 corresponds to no losses of vapors onto wall deposited particles, which provides the lowest bound of SOA yields. As mentioned above, treatments using w=0 or w=1 results in similar yields, which are significantly lower than when considering vapor losses to chamber walls.

"The particle wall loss has already been corrected as described in Stefenelli et al. (2019)."

**11. Line 118: Krechmer et al. (ES&T, 2016) have argued that C\_wall needs to be modeled as a function of C\* based on observations of loss rates of oxidation products with different C\*. Can this also be modeled in this work?**

We are not sure if we understand the reviewer correctly. Krechmer et al. (ES&T, 2016) showed that the modelled C\* is not sensitive to assumption on the assumed  $C_{wall}$  (best seen in their Figure S11 and S12). In our case too,  $C_{wall}$  plays a minor role compared to kw, therefore we kept  $C_{wall}$  as constant in this study

**12. Section 3.2: Perhaps, focus this section on the VBS\_noWLS and VBS\_WLS and frame the other schemes as sensitivity or legacy simulations?**

See answers to the general comments 2. We have added a paragraph and Table 1 to clarify the purpose of each scheme.

**13. Line 162: By how much was the aerosol yield scaled by to account for vapor wall loss?**

In the updated SOAP2 scheme of CAMx, the aerosol yield is parameterized based on new aerosol yield data corrected for vapor wall losses in smog chamber (Zhang et al., 2014; Hodzic et al. 2016). The difference between old (SOAP) and new SOAP (SOAP2) is shown below in Table R1 and R2. However, one should notice that vapor wall loss correction is not the only difference between the datasets used to fit old and new SOAP parameters.

| SOA species | VOC precursor | Aerosol mass yield 1 | C* [µg/m 3 ] at 298K | $\Delta H^{vap}$ [kJ/mol] |
|-------------|---------------|---------------------------------|---------------------------------|---------------------------|
| SOA1        | Benzene       | 0 / 0.605                       | 48                              | 20                        |
|             | Toluene       | 0/0.137                         |                                 |                           |
|             | Xylene        | 0 / 0.093                       |                                 |                           |
|             | IVOC          | 0/0                             |                                 |                           |
| SOA2        | Benzene       | 0 / 0.036                       | 1.6                             | 24                        |
|             | Toluene       | 0 / 0.064                       |                                 |                           |
|             | Xylene        | 0 / 0.036                       |                                 |                           |
|             | IVOC          | 0.224 / 0.200                   |                                 |                           |
| SOPA        | Benzene       | 0.37 / 0.019                    | 0                               | -                         |
|             | Toluene       | 0.30/0                          |                                 |                           |
|             | Xylene        | 0.36 / 0.00006                  |                                 |                           |
|             | IVOC          | 0.348 / 0.183                   |                                 |                           |

Table R1 SOA parameters for SOAP (original, from Ramboll Environ (2016))

 Table R2 SOA parameters for SOAP2 (based on vapor wall loss corrected datasets, from Ramboll (2018))

| SOA species | VOC precursor | Aerosol mass yield 1 | C* [µg/m 3 ] at 300K | ∆H vap [kJ/mol] |
|-------------|---------------|---------------------------------|---------------------------------|----------------------------|
| SOA1        | Benzene       | 0.487 / 0.248                   | 14                              | 116                        |
|             | Toluene       | 0.663 / 0.304                   |                                 |                            |
|             | Xylene        | 0.291 / 0.084                   | ]                               |                            |
|             | IVOC          | 0/0.012                         |                                 |                            |
| SOA2        | Benzene       | 0.167 / 0.391                   | 0.31                            | 147                        |
|             | Toluene       | 0.345 / 0.293                   | ]                               |                            |
|             | Xylene        | 0.306 / 0.049                   |                                 |                            |
|             | IVOC          | 0.275 / 0.225                   |                                 |                            |
| SOPA        | Benzene       | 0/0                             | 0                               | -                          |
|             | Toluene       | 0.262 / 0.044                   |                                 |                            |
|             | Xylene        | 0.294 / 0.025                   |                                 |                            |
|             | IVOC          | 0.277 / 0.129                   | ]                               |                            |

1 Mass-based yields of CG products from VOC precursors (low-NOx yield / high-NOx yield)

14. VBS\_WLS: How does one think about the relatively higher NOx conditions in the chamber experiments and the much lower NOx conditions in the atmosphere while translating the SOA parameters from the chamber to the atmosphere? How can this influence be further studied via sensitivity simulations?

Our study is representative of a high-NOx regime and thus assesses the SOA forming potential for this atmospherically relevant condition. We note that the atmosphere in Europe is generally at high  $NO_x$  conditions.

**15. Lines 202-203: What was the OM loading in the other experiments that is compared to here? The average OM loading for the 14 experiments is 87.8 $\mu$ g m-3, with a range of 19 to 284 $\mu$ g m-3. The OM loads of Exp9 and Exp14 are at relatively low level.**

16. Line 215: Is the temperature effect on gas-phase chemistry taken into account? Yes. Temperature influence both the reaction and partition in the box model.

17. Section 4.1, last paragraph: I think I understand what you mean but try to explain the 'vapor wall loss yield' results better since one would expect the yields to be lower when accounting for vapor loss to the walls.

We updated the statement in Section 4.1 to clarify the two cases (P9, L255–L256).

"The optimized volatility distribution for the secondary condensable gases from biomass burning (ppm per ppm IVOC) based on different wall loss assumptions ( $k_w > 0$  or  $k_w = 0$ ) are displayed in Fig. 3a. The optimized yields considering the vapor wall loss leads to a 3.3 times higher mass in the lowvolatility bins (logC\*  $\leq 0$ ) compared to that assuming  $k_w = 0, ...$ "

**18. Figure 3(a): Are those emission factors or production factors for SOA?**

It refers to ppm of condensable gases generated by per ppm of IVOC emissions. We updated the yaxis-label of Fig. 3a to "Yield factors (ppm/ppm)" to make it clear.

**19. Figure 5: What is the R2? Can you describe the model skill and how it compares across schemes, cities, and other comparable models?**

We added a column in Table 2 for Pearson correlation coefficient (r). The model performance at site scale has actually been already discussed in L280–L293, and comparison of measured and observed OA, POA and SOA across schemes at each site can be found in Fig. S4. It would certainly be

interesting to compare with other models, but we can hardly get the results of other models at the same site and same period, and we think it is somehow beyond the focus of this study.

| Species | OA scheme | MB (µg m -3 ) | ME (µg m -3 ) | RMSE (µg m -3 ) | MFB (%) | MFE (%) | r    |
|---------|-----------|--------------------------|--------------------------|----------------------------|---------|---------|------|
| OA      | SOAP      | -4.1                     | 4.9                      | 7.2                        | -44.3   | 65.3    | 0.38 |
|         | VBS_BASE  | -4.9                     | 5.6                      | 7.9                        | -72.9   | 83.3    | 0.29 |
|         | VBS_3POA  | -1.6                     | 4.3                      | 6.5                        | -12.4   | 51.7    | 0.42 |
|         | VBS_noWLS | -1.9                     | 4.3                      | 6.5                        | -17.4   | 52.7    | 0.41 |
|         | VBS_WLS   | -0.4                     | 4.6                      | 6.9                        | -1.6    | 52.2    | 0.41 |
| SOA     | SOAP      | -2.3                     | 3.1                      | 4.3                        | -77.8   | 98.3    | 0.12 |
|         | VBS_BASE  | -1.6                     | 2.8                      | 4.1                        | -63.0   | 90.6    | 0.22 |
|         | VBS_3POA  | -1.2                     | 2.8                      | 4.1                        | -51.1   | 84.3    | 0.23 |
|         | VBS_noWLS | -1.3                     | 2.8                      | 4.0                        | -52.5   | 84.9    | 0.24 |
|         | VBS_WLS   | 0.2                      | 3.2                      | 4.6                        | -20.0   | 76.4    | 0.26 |
| POA     | SOAP      | -0.7                     | 1.9                      | 3.1                        | 4.4     | 56.7    | 0.49 |
|         | VBS_BASE  | -2.3                     | 2.5                      | 4.0                        | -64.1   | 81.5    | 0.44 |
|         | VBS_3POA  | 0.8                      | 2.4                      | 3.4                        | 36.3    | 64.2    | 0.45 |
|         | VBS_noWLS | 0.4                      | 2.2                      | 3.2                        | 30.1    | 61.9    | 0.45 |
|         | VBS_WLS   | 0.6                      | 2.3                      | 3.3                        | 32.4    | 62.5    | 0.45 |

**Table 2:** Statistical results for model performance on simulating OA, SOA and POA. The number of daily average observations from five ACSM/AMS stations is 216.

20. Figure 6: It is surprising to see so much SOA production compared to POA even in the winter. Are aqueous pathways for SOA production accounted for? Were the chamber experiments done under 'dry' or 'wet' conditions?

According to our discussion in section 4.3.2 (see Table S2), the modeled ratios of SOA to OA are actually even slightly lower than the measurements in winter. Therefore, we believe the SOA production is in a reasonable range compared to POA in this study. In CAMx v6.50, alpha-dicarbonyl compounds which can form secondary organic aerosol (SOA) via aqueous-phase reactions are included in CB6 mechanism, and aqueous formation of SOA from glyoxal, methyl glyoxal and glycoaldehyde is handled by the RADM aqueous chemistry module (Ramboll, 2018). The chamber experiments were conducted under "wet" condition with a relative humidity of 50%.

**21. Figure 9: Is this just for winter or all year round?**

Figure 9 is for the year round due to the limited number of reported measurements. We updated the caption to clarify it.

"*Figure 9:* Comparison between modeled and measured fSOA from literature over the year (see data and sources in Table S2). The shadows are confidence intervals of the regression lines."

22. Line 283: Can you make emissions maps to make this point about the dominance of biomass burning emissions in certain locations clear?

We added the emission maps for the major precursors ( $PM_{2.5}$  and VOC) from biomass burning in 2011 as Fig. S5.

**Figure S5:** Spatial distribution of PM2.5 (left) and NMVOC (right) annual emissions from residential biomass burning in 2011.

**23. Section 4: The paper would benefit from doing a sensitivity simulation where biomass burning is eliminated from the emissions to understand the relative contribution of this source to total OA/POA/SOA.**

We totally agree it is important to understand the relative contribution of biomass burning to the total OA. Actually we have already published a similar work (Jiang et al., 2019) to separate the contribution of biomass burning based on the same inputs but with an earlier version of CAMx (version 6.30). Figure below is produced using a slightly different VBS scheme in CAMx v6.3 (we do not expect dramatic differences compared to this study). The reason we did not include the source apportionment in this study is that there is no BBSOA measurement we can use to validate the model results. We think that simply separating the contribution of biomass burning without evaluation with measurements, would be a duplication of our previous work and would have limited contribution to understand the roles of vapor wall loss correction in modeling – which is the key focus of this study. We added a sentence in Section 4.3.1 (P10, L306–L308) to highlight the contribution of biomass burning to POA and SOA.

"...Among all the sources, residential biomass burning contributed to 16.3–52.6% of POA and 5.9-28.9% of SOA in winter (Jiang et al., 2019b), indicating the potential roles of vapor wall loss for the biomass burning sector..."

---

## Author Comment (AC2) · 15 Dec 2020

**Responses to the comments of anonymous referee #2**

We thank the referee for the valuable comments that have greatly helped us to improve the manuscript. Please find below our responses (in black) after the referee comments (in blue). The changes in the revised manuscript are written in *italic*.

General Comments Jiang et al. report a modeling study that: (1) evaluates the impact of vapor wall losses during chamber studies on parameters for SOA formation from residential biomass burning emissions, and (2) simulates the increases in OA and SOA concentrations once a CTM is updated with new SOA parameters that take into account these wall losses. Overall, the manuscript is well-written and addresses an important topic in the field of atmospheric aerosol modeling, since implementing accurate parameters for SOA formation in CTMs is challenging. I support publication in GMD once my comments below have been taken into consideration.

Line-by-line comments
Page 1, Line 27 – 29: What is the difference between the "standard VBS" and the "reference scenario"? Does the reference scenario refer to the traditional two-product approach? Please clarify as this is an interesting finding.
The "standard VBS (VBS_BASE)" refers to the default VBS parameterization in CAMx v6.5, while in the "reference scenario (VBS_noWLS)" and "wall loss corrected scenario (VBS_WLS)" we adopted the optimized parameterizations based on the chamber experiments with different assumptions: there is no loss of condensable gases to the chamber wall for VBS_noWLS, and there is for VBS_WLS. We updated the abstract to clarify it. A paragraph as well as a Table 1 were added to Section 3.2.1 to describe the difference of each OA schemes.

*"The modeled results from the VBS schemes with standard (VBS_BASE) and vapor wall loss corrected parameters (VBS_WLS), as well as the traditional two-product approach were compared and evaluated by OA measurements from five Aerodyne aerosol chemical speciation monitor (ACSM)/aerosol mass spectrometer (AMS) stations in the winter of 2011. An additional reference scenario VBS_noWLS was also developed using the same parameterization as VBS_WLS except for the SOA yields which was optimized assuming there is no vapor wall loss. The VBS_WLS generally shows the best performance for predicting OA among all OA schemes, and reduces the mean fractional bias from -72.9% (VBS_BASE) to -1.6% for the winter OA. In Europe, the VBS_WLS produces the highest domain average OA in winter (2.3 $\mu g\ m^{-3}$), which is 106.6% and 26.2% higher than VBS_BASE and VBS_noWLS, respectively." (P1, L21–29)*

*"To investigate the effects of vapor wall loss corrected yields, as well as to compare to other modifications/parameterizations that are currently strongly debated in the community, five simulations with different OA schemes were conducted in this study (Table 1). Besides VBS_WLS which uses the optimized parameterization with vapor wall loss correction for the biomass burning sector, SOAP and VBS_BASE represent the two standard parameterization in CAMx; VBS_3POA represents a common approach to offset the missing SVOC emissions in recent modelling studies without vapor wall loss; VBS_noWLS is another reference case for that without vapor wall loss, which uses exactly the same parameters as VBS_WLS except for the SOA yields from IVOCs. Details about each OA schemes are introduced below:..." (P6, L173–179)*

**Table 1:** Description about the different OA schemes.

| OA scheme | IVOB[a] emissions | $k_{OH}$ for IVOB (cm$^3$ molec$^{-1}$ s$^{-1}$) | SOA yields for IVOB (ppm/ppm)[b] |
|---|---|---|---|
| SOAP | = 4.5*POA_BB | 1.34 | /[c] |
| VBS_BASE | | 4.0 | [0.081, 0.135, 0.800, 0.604, 0.0] |
| VBS_3POA | | 4.0 | [0.081, 0.135, 0.800, 0.604, 0.0] |
| VBS_noWLS | = 12*POA_BB | 1.5 | [0.014, 0.036, 0.076, 0.136, 0.44] |
| VBS_WLS | | 1.5 | [0.078, 0.118, 0.157, 0.177, 0.312] |

[a] IVOB is the abbreviation of "IVOC from Biomass Burning" in CAMx

[b] The yield values are corresponding to volatility bins with saturation concentrations of $10^{-1}$, $10^0$, $10^1$, $10^2$ and $10^3$ µg m$^{-3}$.

[c] SOAP does not separate IVOC from biomass burning and other anthropogenic sectors, and therefore is not comparable with the SOA yields for IVOBs.

Less importantly, I also don't understand why the authors have specifically highlighted the result from Romania.
We rephrased the sentence as *"VBS_WLS leads to an increase in SOA by up to ~80% (in Balkans)"* (P1 L30)

Page 2, Line 37 – 39: The authors imply that residential biomass burning emission is "the dominant source for. . . secondary organic aerosols in winter". However, this statement is supported exclusively by three European studies that are referenced on line 39. Therefore, the authors need to clarify that this conclusion is specific to the European domain.
We rephrased the sentence (in P2, L39) to clarify it refers to the European domain.

*"...residential biomass burning emissions have been recognized as the dominant source for both primary (POA) and secondary (SOA) organic aerosols in Europe during winter time"*

Page 2, Line 47: I would kindly suggest that the authors specify that the POA emissions are treated as semi-volatile when this "scaling-up" is performed, since some models still assume that POA is nonvolatile.
Done. We updated the sentence as follows:

*"In order to compensate the effects from missing precursors, various modeling studies treated the POA as semi-volatile and increased the POA emissions by a factor based on findings from chamber experiments…"* (P2, L49)

Page 2, Line 57: The work of Hayes et al. 2015 concerning vapor wall losses used a box model and not a CTM.
We apologize for the mistake. The statements are now corrected in P2, L60.

*"This factor was also implemented in a box model with volatility basis set (VBS) scheme (Hayes et al., 2015),…"*

Section 2.1: How does the utilization of beech wood as the only fuel potentially bias the results? Is this a fuel commonly used in residential biomass burning in Europe? Basing the parameterization of the VBS scheme on a single fuel is not necessarily a flaw in the study, but some contextualization is needed here to understand how this limitation might influence the model results.
Beech is one of the major forest trees in Europe, and beech wood is widely used for combustion as a heating fuel in European households. That's why beech wood is selected in our chamber experiments.

Different biomass fuel types may influence the vapor composition. Nevertheless, a recent study showed that despite large differences in fuel type and burning conditions, SOA formed is consistent for similar total NMOG load and OH exposure (Lim et al., 2019). This is also consistent with our results showing no dependence of the closure on the combustion regime (flaming vs. smoldering). Therefore, while we think that the fuel type may have an influence on the results this influence is minor compared to other sources of uncertainties (e.g. vapor wall losses or not considering the influence of SVOC on SOA formation). We added the explanation in P3, L83–L87.

*"Beech wood is selected as it is one of the major forest types in Europe, and is widely used for residential heating and cooking in Europe. Although different biomass fuel types may largely affect the emitted organic gas species and affect the SOA formation, a recent study showed that the effect of biomass fuel type on SOA formation is much smaller than the effects of initial OM load and OH exposure (Lim et al., 2019)"*

Line 105: The phrase "gas-phase equilibrium concentrations in particle phase" is not coherent. Please clarify. Furthermore, I don't think equation (3) can be correct. A partitioning coefficient of 1 would indicate complete partitioning to the particle phase, but this would give a $C_{eq}$ (i,p) value of zero. This comment also applies to equation (4).

Here is an ambiguity in the term we have used. The $C_{eq}$ refers to gas-phase concentrations at equilibrium with respect to the particle phase ($C_{eq,p}$) and to the chamber walls ($C_{eq,w}$). Therefore, a partitioning coefficient of 1 would indicate complete partitioning to the particle phase, which is consistent with a gas-phase concentration $C_{eq}$ (i,p) of zero. We updated the explanation of $C_{eq}$ in P4, L122–L123.

*"the gas-phase concentrations at equilibrium with respect to the particle phase ($Ceq_{i,p}$) and to the chamber wall ($Ceq_{i,w}$)"*

Line 201: The reasoning why the OM loading would have an effect on the box model's accuracy is not clear. Please elaborate. Also, there are some runs when the are low when the model accuracy is very reasonable, for example a11, so the OM loading does not seem to explain by itself the poor accuracy of the model observed for experiments 9 and 14.

Since we optimized the parameters by the sum of mean bias (MB) and RMSE between modelled and measured OA for all 14 experiments, experiments with higher OM loads and SOA productions (which normally have larger MB and RMSE), have higher impact during the model optimization, leading to a better model performance for experiments with higher OM loads after optimization. That is why we mentioned that experiments with lower OM loads may have lower accuracy. We clarify this in the new version of the manuscript.

*"The model reproduces the process of OA formation for most of the experiments well, except for experiment #9 and #14 which have relatively lower OM loads (26 and 48 µg m⁻³ for Exp9 and Exp14, respectively). It can be partially explained by different weighting impact for experiments with high or low OM loads. The experiments with higher OM loads normally have larger MB and RMSE in the beginning of optimization, and therefore have higher impact during the model optimization. A direct consequence is the optimized parameters would work better for those experiments with higher OM loads. However, the model performance on each experiment could also be influenced by a series of other factors such as temperature and chamber conditions…"* (P8, L230–L234)

Figure 1: It would be useful if a table of the experiment conditions was provided.

We did not describe the experimental conditions in detail as they have already been reported in our previous publication (Stefenelli et al., 2019). As suggested by the referee, we added additional information in Table S1.

*"The conditions of each chamber experiment are shown in Table S1. More detailed description of the experiments can be found in Stefenelli et al. (2019), Bertrand et al. (2017) and Bruns et al. (2016)."* (P3, L90)

**Table S1:** Experimental conditions for the 14 chamber experiments used in this study.

| #Exp | References | Date | Experimental temperature (°C) | Stove type [a] | OM loads ($\mu$g m$^{-3}$) |
|---|---|---|---|---|---|
| 1 | Bertrand et al. (2017) | 29.10.2015 | 2 | stove 1 | 198 |
| 2 | Bertrand et al. (2017) | 30.10.2015 | 2 | stove 1 | 285 |
| 3 | Bertrand et al. (2017) | 04.11.2015 | 2 | stove 1 | 123 |
| 4 | Bertrand et al. (2017) | 05.11.2015 | 2 | stove 1 | 46 |
| 5 | Bertrand et al. (2017) | 06.11.2015 | 2 | stove 2 | 75 |
| 6 | Bertrand et al. (2017) | 07.11.2015 | 2 | stove 2 | 134 |
| 7 | Bertrand et al. (2017) | 09.11.2015 | 2 | stove 2 | 81 |
| 8 | Bruns et al. (2016) | 02.04.2014 | -10 | stove 3 | 19 |
| 9 | Bruns et al. (2016) | 17.03.2014 | -10 | stove 3 | 26 |
| 10 | Bruns et al. (2016) | 25.03.2014 | 15 | stove 3 | 62 |
| 11 | Bruns et al. (2016) | 27.03.2014 | 15 | stove 3 | 45 |
| 12 | Bruns et al. (2016) | 28.03.2014 | 15 | stove 3 | 42 |
| 13 | Bruns et al. (2016) | 29.03.2014 | 15 | stove 3 | 48 |
| 14 | Bruns et al. (2016) | 30.03.2014 | 15 | stove 3 | 48 |

[a] Stove 1 manufactured before 2002 (Cheminées Gaudin Ecochauff 625), stove 2 fabricated in 2010 (Invicta Remilly) and stove 3 (Avant, 2009, Attika).

Line 203 – 204: I think the text contains an error here. If anything there is an underestimation at short times and an overestimation at long times. More generally, it seems like the comparison between the model and the measurement varies a lot between experiments, so it is difficult to make conclusions regarding whether the box model is overestimating or underestimating.

We believe there is some misunderstanding here. We do not mean to conclude whether the model is overestimating or underestimating here, but to show the agreement between the shapes of measured and modeled curves is improved for most of the experiments when the vapor wall loss is considered. We clarify the point in P8, L235–L236

*"... the agreement between the modeled and measured trends was improved when the vapor wall loss is taken into account."*

Lines 205 – 207: I think using percentages here to compare the two box model versions (with or without vapor wall losses) overstates the difference between the models. In the end, the differences in the MB and RMSE are only about 6 ug/m3, which is not very much when most of the experiments are run at OM concentrations near or above 100 ug/m3.

The average mean bias was improved from -12.8 to 6.7 $\mu$g m$^{-3}$. The change from underestimation to overestimation somehow indicates we find a potential reason for the long-existing problem – underestimated OA in modeling studies, which means more than the absolute change of 6 $\mu$g m$^{-3}$. In addition, the model performance is not only about the bias at the end point, but also the capability to reproduce the chemical processes of OA formation. As we mentioned before, the shape of the

modeled curves for OA formation with vapor wall loss corrections shows higher agreement with the measurements.

Figure 3: I very strongly suggest that these data also be given in a table so that the quantitative results can be used by other researchers.
We will upload the data for each figure after the acceptance of the manuscript.

Lines 284 – 287: This sentence is confusing. Which model cases are specifically being compared? In addition, in Figure 7, only the schemes VBS_WLS and VBS_noWLS are compared, but then in the text the 3POA scheme is mentioned as well.
We updated the sentence in L284 (current L316) as follows to make it clear.

"*The overall relative differences between VBS_WLS and VBS_noWLS are more than 80% and the highest grid-scale increment reaches 5.6 µg m$^{-3}$ in the region of Balkans.*"

We are sorry for the confusion for Fig. 7. The "Fig. 7" here is a typo, and we have corrected it to "Fig. 6" (P11, L320).

Figure 8: It would be helpful to specify in the figure caption that these plots are annual averages. In addition, why are annual averages used and discussed rather than wintertime measurements, as is done in the other sections of the manuscript?
We specified it is the annual average in the caption of Fig. 8 as the referee suggested. We compared the modeled SOA/OA fraction (fSOA) with the measurements from literature covering different seasons in Section 4.3.2. The number of reported measurements is limited if only winter data were included. That's why we discussed the whole year in this section.

"***Figure 8:*** *Modeled fractions of annual mean SOA to total OA (fSOA) using different OA schemes. Modeled results of VBS_3POA are very similar to VBS_noWLS, and therefore are not shown here.*"

Lines 318 – 322: The comparisons summarized here are rather haphazard. First, for OA, the VBS_WLS scheme is compared to the VBS_BASE scheme. Then next, for SOA, the VBS_WLS scheme is compared to the SOAP scheme. The authors should be consistent in what schemes they are comparing to as "base cases".
Here we intend to show the ranges of improvement. We mentioned the VBS_BASE and SOAP specifically because they had the largest absolute MFB for OA and SOA, respectively. We updated the sentence (P12, L354) to specify they refer to the largest mean fractional bias.

"*Comparison of the modeled results with different OA schemes with the field measurements from five ACSM/AMS stations in Europe in winter, suggests that VBS_WLS generally has the best performance to predict OA, which lowers the highest mean fractional bias from -72.9% (VBS_BASE) to -1.6% for OA, and -77.8% (SOAP) to 20.0% for SOA.*"

**References**
Lim, C. Y., Hagan, D. H., Coggon, M. M., Koss, A. R., Sekimoto, K., de Gouw, J., Warneke, C., Cappa, C. D., and Kroll, J. H.: Secondary organic aerosol formation from the laboratory oxidation of biomass burning emissions, Atmos. Chem. Phys., 19, 12797-12809, 10.5194/acp-19-12797-2019, 2019.
Stefenelli, G., Jiang, J., Bertrand, A., Bruns, E. A., Pieber, S. M., Baltensperger, U., Marchand, N., Aksoyoglu, S., Prévôt, A. S. H., Slowik, J. G., and El Haddad, I.: Secondary organic aerosol formation from smoldering and flaming combustion of biomass: a box model parametrization based on volatility basis set, Atmos. Chem. Phys., 19, 11461-11484, 10.5194/acp-19-11461-2019, 2019.

---

## Author Response (AR2)

**Responses to the comments of anonymous referee #1**

We thank the referee for the additional comments. Please find below our responses (in black) after the referee comments (in blue). The changes in the revised manuscript are written in *italic*.

The authors seem to have responded to both sets of comments and questions posed by the two reviewers. I have specific concerns on a few of the responses (see below) but should note that most of my concerns are centered around the description of the methods (and not the results). I am generally okay with the response and changes made to the manuscript and recommend publication of this paper in GMD after the authors have had a chance to review my additional concerns below.

**Continuing comments:**
1 Reviewer 1, point 1:
a.  Thanks for providing additional details on the PTR data but I still think the paper would benefit from a brief description of the PTR measurements, speciation information, and how those species were aggregated for use in the model, including the rationale (e.g., reduced-form?) and limitations (e.g., highly lumped?) of this approach. This detail is limiting comprehension of the modeling sections in 2.2 and 2.3 and could be done in the SI if there are issues with space.
We added a brief description about the measurements and the approach to group the species in section 2.1 and section 2.2.
*"The PTR-ToF-MS was operated under standard conditions in $H_3O^+$ mode, as introduced in Stefenelli et al. (2019). A common set of 263 ions was extracted from the measurements, and among these ions, 86 showed clear decay with time and were identified as potential SOA precursors. These are listed in Table S1 of Stefenelli et al. (2019)."* (P3, L90–L93, section 2.1)
*"CAMx includes four types of precursors from anthropogenic sources, i.e. toluene, xylene, benzene, and IVOC which includes all the other unspeciated organic gases. According to our measurements, the traditional anthropogenic precursors toluene, xylene and benzene only account for ~15% of the total organic gases. To facilitate the implementation of the optimized parameters in CAMx, all the measured SOA precursors including the traditional ones were lumped into one surrogate as IVOC with the same reaction rate and volatility distribution. In comparison, Stefenelli et al. (2019) have assigned the same set of compounds to six different classes according to their properties (reaction rates, expected SOA yields…) and origins/occurrence in the emissions. These included furans and methoxy-phenols from the pyrolysis of cellulose and lignin, respectively, single-ring and poly-aromatic hydrocarbons from flaming combustion, and oxygenated non-aromatic compounds with lower and higher than six carbon chains. The current lumping approach of all these species into one surrogate, despite variations in their properties is more adapted for the implementation into CAMx and for assessing vapor wall losses, where additional parameters are included in the box model."* (P4, L100–L111, section 2.2)

b.  Can you show how the measurements were used to calculate the OH reaction rate constant?

Is there a citation for this calculation?

The $k_{OH}$ was calculated following the Eq (1) in Stefenelli et al. (2019) shown as follows, where P represents the production of oxidized condensable gases (OG) in the chamber, $k_{dil}$ is the dilution rate, and $k_{other}$ is the loss rate of OG by other processes.

$$\frac{d[OG]}{dt} = P - \left(\sum k_{dil} \times [OG] + k_{OH} \times [OH] \times [OG] + k_{other} \times [OG]\right).$$

We added the citation in P7, L217–L218.

*"The reaction rate with OH ($k_{OH}$) was calculated based on the measurements following Stefenelli et al. (2019)"*

2. Reviewer 1, point 2: I still maintain that determining SVOC and IVOC emissions from POA emissions (that are susceptible to vagaries of partitioning) is a poor choice. It should be fine for this work but needs to be discussed in light of arguments made in earlier work, e.g., Lu et al. (2018; https://doi.org/10.5194/acp-18-17637-2018).

More discussion about the S/IVOC calculations are added in the introduction section.

*"In order to compensate the effects from missing precursors, various modeling studies treated the POA as semi-volatile and adopted different scaling approaches to calculate the S/IVOC emissions. The most commonly used method is to increase the POA emissions by a factor of 3 (Ciarelli et al., 2017a; Fountoukis et al., 2014; Jiang et al., 2019b; Tsimpidi et al., 2010), while recent studies have also developed new profiles based on the nonmethane organic compounds (NMOCs) (Lu et al., 2018; Cai et al., 2019). However, increasing number of laboratory experimental studies found that the S/IVOC emissions are of high variability depending on different burning conditions and fuel types (Hatch et al., 2015; Hatch et al., 2017; Jen et al., 2019; Koss et al., 2018; Sekimoto et al., 2018), and the estimation of S/IVOC in modeling studies remains to be improved."* (P2, L46–L53)

...

*"Despite considerable variability of the SVOC emissions from biomass burning according to recent studies, the VBS_3POA is supposed to be a reference case representing the commonly used approach without vapor wall loss, and therefore we adopted the routine approach of multiplying the POA emissions by a factor of 3 to offset the influence of missing SVOC emissions."* (P7, L206–L209)

Lu, Q., Zhao, Y., and Robinson, A. L.: Comprehensive organic emission profiles for gasoline, diesel, and gas-turbine engines including intermediate and semi-volatile organic compound emissions, Atmos. Chem. Phys., 18, 17637-17654, 10.5194/acp-18-17637-2018, 2018.

Cai, S., Zhu, L., Wang, S., Wisthaler, A., Li, Q., Jiang, J., and Hao, J.: Time-Resolved Intermediate-Volatility and Semivolatile Organic Compound Emissions from Household Coal Combustion in Northern China, Environ. Sci. Technol., 53, 9269-9278, 10.1021/acs.est.9b00734, 2019.

3. Why is the sum of the molar yields in Table 1 for IVOCs larger than 1?

We have constrained the total mass yields in the volatility range X-Y to be equal 1 (using normal distribution kernel functions). This does not have a great effect on the resulting yields as the

normal distribution center and width are allowed to vary/adjust. The sum becomes larger than 1 when converting the mass yields to molar yields. The default molar yields in CAMx also have a sum larger than 1 as the VBS scheme accounts for both oxygenation and fragmentation (Koo et al., 2014). We added a sentence in P7, L215–L217 to clarify it.

*"The optimized mass yields in the box model were converted to molar yields using the default molecular weights in CAMx (Table 1). Both the optimized and default molar yields have a sum larger than 1 as the VBS scheme accounts for both oxygenation and fragmentation (Koo et al., 2014)."*

Koo, B., Knipping, E., and Yarwood, G.: 1.5-Dimensional volatility basis set approach for modeling organic aerosol in CAMx and CMAQ, Atmos. Environ., 95, 158-164, 10.1016/j.atmosenv.2014.06.031, 2014.

4. Reviewer 1, point 2: If I understand this correctly, the model assumes that there is only a single IVOC-like precursor that reacts with OH to form biomass burning SOA and that all other precursors (e.g., single-ring aromatics) are ignored. When the IVOC emissions are determined and adjusted to remove other more traditional SOA precursors in CAMx, a few assumptions are being made but none are explained. It assumes that benzene, toluene, and xylenes are the only other traditional SOA precursors. What about larger aromatics, isoprene, monoterpenes, and the likes? It assumes that the yields of the now-removed traditional SOA precursors are identical to the IVOC-like SOA precursor that is being added. On a related note, why weren't the traditional precursors explicitly accounted for in the box model so parameters specific to the IVOC-like SOA precursor could be directly determined and applied in CAMx?

As we mentioned in the response to point 2, CAMx has benzene, toluene, xylene and IVOC as the traditional SOA precursor species from **anthropogenic** emissions. Isoprene and monoterpenes are included in CAMx but mainly from the biogenic sources. We do not find them in our biomass burning emissions. As the modification of parameters in this study is on the biomass burning sector, the biogenic precursors were not specifically mentioned in the method section. We rephrased the sentence to clarify that these are traditional precursors for biomass burning in P7, L220–L221.

*"... among which the traditional precursors in CAMx from biomass burning (toluene, xylene and benzene) accounting for ~15% of the total emissions."*

Separating the traditional precursors and the IVOCs in the box model would mean that we need to determine their yield, which increases the parameter space substantially. Therefore, we decided to lump all the species into one surrogate, IVOC, in the box model and optimize their yield parameters. We have then implemented these parameters in CAMx for biomass burning emissions and excluded the traditional precursors. We modified P4, L100–L111 to better explain that the traditional precursors were not separated in the box model.

*"CAMx includes four types of precursors from anthropogenic sources, i.e. toluene, xylene, benzene, and IVOC which includes all the other unspeciated organic gases. According to our measurements, the traditional anthropogenic precursors toluene, xylene and benzene only account for ~15% of the total organic gases. To facilitate the implementation of the optimized parameters in CAMx, all the measured SOA precursors including the traditional ones were*

*lumped into one surrogate as IVOC with the same reaction rate and volatility distribution. In comparison, Stefenelli et al. (2019) have assigned the same set of compounds to six different classes according to their properties (reaction rates, expected SOA yields…) and origins/occurrence in the emissions. These included furans and methoxy-phenols from the pyrolysis of cellulose and lignin, respectively, single-ring and poly-aromatic hydrocarbons from flaming combustion, and oxygenated non-aromatic compounds with lower and higher than six carbon chains. The current lumping approach of all these species into one surrogate, despite variations in their properties is more adapted for the implementation into CAMx and for assessing vapor wall losses, where additional parameters are included in the box model."*

5. Reviewer 1, point 8: Sinha et al. (2018) heated the aerosol and heating itself could change the mass accommodation coefficient through changes in the viscosity/bulk diffusion coefficient. So, I don't see that study as directly refuting the findings from Cappa and Akherati that used isothermal dilution to probe changes in POA.

As mentioned in our earlier response, by considering the POA volatility distributions from May et al. (2013) our results indicate little evaporation of the POA (~20%), which is in line with Cappa and Akherati. We cannot completely exclude kinetic limitations. Considering lower accommodation coefficients would increase the wall loss effects and would result into higher SOA yields. Therefore, our results using accommodation coefficient = 1 are lowest estimates of the wall loss effects and SOA yields. We have added these considerations in the modified version of the manuscript.

6. Reviewer 1, point 9: The different k_w seem to scale inversely with chamber size. This fact could be mentioned.

We added a sentence about the $k_w$ and chamber size in P4, L127–L128.
*"The $k_w$ varies significantly depending on the chamber conditions such as the chamber size, relative humidity, etc."*

7. Reviewer 1, point 11: See Figure 4 and Figure S10 in Krechmer et al. (2016).

In Figure 4 and S10, Krechmer et al. (2016) shows a dependence of the particle accommodation coefficients, alpha, not $C_w$ on C*. They have mentioned in the text that their $C_w$ has been parameterized in a previous work based on the decay of compounds with different C*, therefore is not independent from C* in their analysis. We note that this dependence of alpha on C* is not large (within 50%) compared to other sources of uncertainties, with alpha < 1 for more volatile compounds with higher C*. Krechmer et al. (2016) noted that the reason for this dependence is not well understood and might be related to effects that were not considered in their model. Therefore, we have not taken this dependence into account in our analysis. We have considered alpha = 1 and as mentioned above this would result in a lower estimate of the vapor wall loss effect and thereby of SOA yields.